



Earth System
Science
Data

# Multi-site, multi-crop measurements in the soil–vegetation–atmosphere continuum: a comprehensive dataset from two climatically contrasting regions in southwestern Germany for the period 2009–2018 [TS1]

**Tobias K. D. Weber**[1]**, Joachim Ingwersen**[1]**, Petra Högy**[2]**, Arne Poyda**[1,a]**, Hans-Dieter Wizemann**[3]**, Michael Scott Demyan**[3,4,b]**, Kristina Bohm**[1,i]**, Ravshan Eshonkulov**[1,c]**, Sebastian Gayler**[1]**, Pascal Kremer**[1]**, Moritz Laub**[3]**, Yvonne Funkiun Nkwain**[2]**, Christian Troost**[4]**, Irene Witte**[1]**, Georg Cadisch**[4]**, Torsten Müller**[5]**, Andreas Fangmeier**[2]**, Volker Wulfmeyer**[3]**, and Thilo Streck**[1]

[1]Institute of Soil Science and Land Evaluation, University of Hohenheim, Stuttgart, Germany
[2]Institute of Landscape and Plant Ecology, University of Hohenheim, Stuttgart, Germany
[3]Institute of Physics and Meteorology, University of Hohenheim, Stuttgart, Germany
[4]Institute of Agricultural Sciences in the Tropics, University of Hohenheim, Stuttgart, Germany
[5]Institute of Crop Science, University of Hohenheim, Stuttgart, Germany
[a]current address: Grass and Forage Science/Organic Agriculture, [CE1] Institute of Crop Science and Plant Breeding, Kiel University, Germany
[b]current address: School of Environment and Natural Resources, The Ohio State University, Columbus, Ohio, USA
[c]current address: Karshi Engineering Economics Institute [CE2], Karshi, Uzbekistan
[i]previously published under the name Kristina Imukova

**Correspondence:** Tobias K. D. Weber (tobias.weber@uni-hohenheim.de)
and Joachim Ingwersen (jingwer@uni-hohenheim.de)

**Abstract.** [CE3]We present a comprehensive, high-quality dataset characterizing soil–vegetation and land surface processes from continuous measurements conducted in two climatically contrasting study regions in southwestern Germany: the warmer and drier Kraichgau region with a mean temperature of 9.7 °C and annual precipitation of 890 mm and the cooler and wetter Swabian Alb with mean temperature 7.5 °C and annual precipitation of 1042 mm. In each region, measurements were conducted over a time period of nine cropping seasons from 2009 to 2018. The backbone of the investigation was formed by six eddy-covariance (EC) stations which measured fluxes of water, energy and carbon dioxide between the land surface and the atmosphere at half-hourly resolution. This resulted in a dataset containing measurements from a total of 54 site years [CE4] containing observations with a multitude of crops, as well as considerable variation in local growing-season climates.

The presented multi-site, multi-year dataset is composed of crop-related data on phenological development stages, canopy height, leaf area index, vegetative and generative biomass, and their respective carbon and nitrogen content. Time series of soil temperature and soil water content were monitored with 30 min resolution at various points in the soil profile, including ground heat fluxes. Moreover, more than 1200 soil samples were taken

to study changes of carbon and nitrogen contents. The dataset is available at https://doi.org/10.20387/bonares-a0qc-46jc (Weber et al., 2021). One field in each region is still fully set up as continuous observatories for state variables and fluxes in intensively managed agricultural fields.

## 1   Introduction

It is well acknowledged that interactions between the soil–vegetation system and the atmosphere will have major impacts on regional climate and that our knowledge of processes and feedbacks is insufficient (Pielke et al., 2007; Thornton et al., 2014). Process models enable testing of hypotheses concerning the governing processes, identifying epistemic and aleatory uncertainties and highlighting the need for further investigations (Porter and Semenov, 2005; Godfray et al., 2010; Challinor et al., 2014; Tao et al., 2017; Schalge et al., 2020). Predicting the impacts of climate change on agro-ecosystems and the land surface exchange of water, energy and momentum and vice versa requires process models to understand and study land–atmosphere feedbacks (Ingwersen et al., 2018; Monier et al., 2018). There is consensus that fully coupled climate, land surface, crop and hydrological models facilitate the prediction of climate change impacts on agricultural productivity as well as its feedbacks on climate change projections themselves (Marland et al., 2003; Hansen, 2005; Perarnaud et al., 2005; Levis, 2010). This implies the continuous improvement of models and process understanding. In relation to the water balance this includes, in particular, partitioning evaporation and transpiration (Kool et al., 2014; Stoy et al., 2019), modelling crop transpiration (Heinlein et al., 2017), investigating impacts on groundwater resources (Riedel and Weber, 2020), improving the representation of the green-vegetation-fraction dynamics of croplands in the Noah-Multiparameterization Land Surface Model (NOAH-MP LSM; Imukova et al., 2015; Bohm et al., 2019), determining the dynamic root growth of crops (Gayler et al., 2014) and assessing the relevance of subsurface processes (Gayler et al., 2013), and evaluating the energy balance closure problem in eddy-covariance (EC) measurements (Ingwersen et al., 2015; Imukova et al., 2016) and associated minor storage terms (Eshonkulov et al., 2019), as well as incorporating crop growth in land surface models (Ingwersen et al., 2011, 2018), investigating the carbon balance and turnover of agro-ecosystems (Demyan et al., 2016; Poyda et al., 2019), evaluating crop model performances (Bassu et al., 2014; Kimball et al., 2019), responding to changes in environmental drivers (Biernath et al., 2011, 2013), quantifying the effect of different intensities of free-air carbon dioxide and temperatures on grain yield and grain quality (Högy et al., 2010, 2019), evaluating the worth of observed data (Wöhling et al., 2013b), and developing data model integration techniques (Wöhling et al., 2013a).

However, the effects are further reaching than just to the biophysical environment. Regional climate projections typically neglect changes and adaptation of the agents of land use, namely the farmers, meaning that the concomitant projections of future crop yields are based on crude simplifications (Hermans et al., 2010). Multi-agent system modelling has reached a level of maturity such that empirical bio-economic simulators can be run on high-performance computer clusters (Schreinemachers and Berger, 2011; Kelly et al., 2013). As a result, integrated model systems (Fig. 1) can now be built that simulate both biophysical and socioeconomic processes with comparable process detail, accounting for the complex reality of local/regional human adaptation and feedback to global changes (Troost and Berger, 2015).

To enable an understanding of feedbacks within bio-economic modelling systems, the models employed for the representation of processes of different complexity in the soil–vegetation–atmosphere continuum require calibration and validation against observed state variables or fluxes at the field level (Kersebaum et al., 2015). For this, high-quality observed data on the state variables or fluxes of interest are required, which should encompass grain and biomass yields and soil organic carbon and nitrogen stocks and turnover in soils, as well as the water, carbon dioxide and energy fluxes between land surface and atmosphere. Still very few model intercomparison studies include, in addition to crop growth, soil water flux relevant variables to calibrate their agro-ecosystem models (Seidel et al., 2018) because datasets that include all these variables and fluxes are rare (Kersebaum et al., 2015). The dataset presented here is intended to help close this data gap, leading to better process representation on the one hand, while, on the other hand, facilitating model selection (Wöhling et al., 2015) and tackling the question of required and sufficient model complexity in the light of available data (Guthke, 2017).

To study the effects of regional climate change and to facilitate parameterization and validation to continuously improve model components, extensive collaborative field measurements and controlled exposure experiments were carried out in two study areas in southwestern Germany. Field research was part of two wider integrated research projects funded by the German Research Foundation (DFG), PAK CE5 346 Structure and Function of Agricultural Landscapes under Global Climate Change – Processes and Projections on a Regional Scale and Research Unit (RU) 1695 Agricultural Landscapes under Global Climate Change – Processes and Feedbacks on a Regional Scale.

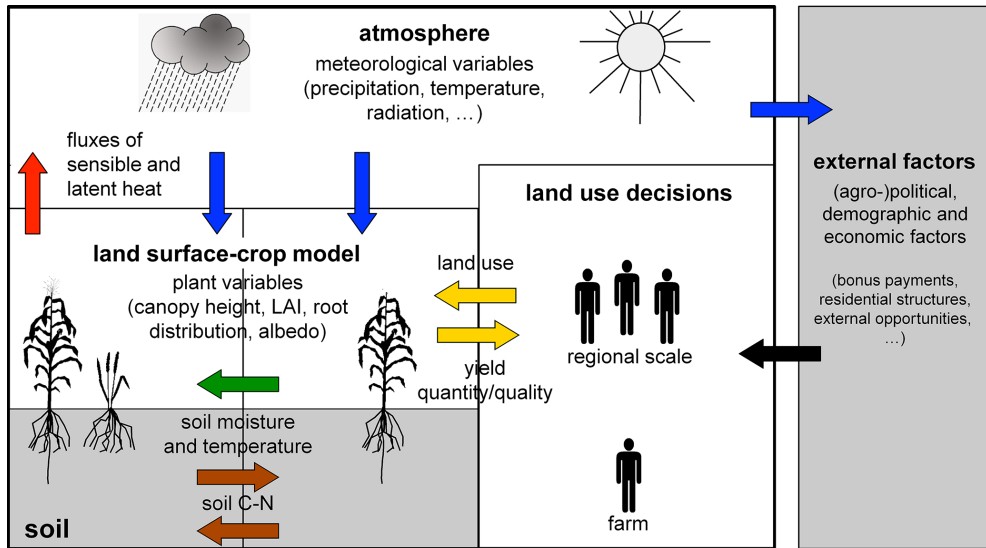

**Figure 1.** Diagram of the cardinal land modelling system compartments and relations. The presented dataset contains time series of quantified land surface, crops, and soil processes and properties. This serves as a unique backbone for model validation and model development in the soil–vegetation–atmosphere continuum and robust land systems modelling.

## 2 Material and methods

In this section, the full dataset, which is composed of many individual datasets spanning diverse types of data sources, temporal and spatial measurement resolution, and origins, is individually described. Both research areas were intensively used agricultural landscapes: (1) Kraichgau, with a mild climate and moderate precipitation and which is dominated by intensive row crop agriculture, and (2) the Central Swabian Alb (Mittlere Schwäbische Alb), with a harsh climate and higher precipitation. Animal fattening, row crop agriculture and heathland areas are important features to the Central Swabian Alb agro-economic setting. Within the scope of this publication we present a high-quality dataset spanning a time period of nine cropping seasons from 2009 to 2018 intensely characterizing the two respective agro-ecosystems. The backbone of the investigation was formed by six eddy-covariance stations which measured fluxes of water, energy and carbon dioxide between the land surface and the atmosphere at half-hourly resolution. This resulted in a dataset containing measurements from a total of 54 site years (i.e. 2 regions × 3 fields × 9 cropping seasons) containing observations with a multitude of crops, as well as considerable variation in local growing-season climates. A detailed graphical schema describing the measurement campaign is presented in Fig. 2. The dataset comprises (i) soil profile characteristics; (ii) management and cultivation data including sowing date, harvest date, crop type and variety, fertilization and pesticide application including amount and type with 1–4 observations $yr^{-1}$, and soil tillage; (iii) meteorological data at 30 min resolution comprising rain, air temperature at 2 m height, relative humidity, and wind direc-

tion and speed; (iv) soil–/biosphere–atmosphere fluxes using fully equipped eddy-covariance stations for carbon, energy and water vapour flux measurements, as well as wind speed and wind direction (from 2009 to 2016 fluxes were not measured during the winter months); (v) soil state measurements including water content, temperature and matric potential (30 min), soil profile depth permitting at 5, 15, 45, 75, 90 and 130 cm soil depth; (vi) CE6 carbon and nitrogen measurements integrated over depths of 0–30, 30–60 and 60–90 cm (4–6 observations $yr^{-1}$); and (vii) plant performance including phenology, height and leaf area index with an average frequency of 7 observations $yr^{-1}$; yield with 1 observation $yr^{-1}$; aboveground biomass with 3–5 observations $yr^{-1}$; and carbon and nitrogen in vegetative – sometimes separated in different plant compartments – and generative biomass. In addition, in selected years and at selected sites, microbial carbon and nitrogen contents and $CO_2$ fluxes between the soil and the atmosphere were determined on vegetated and bare-soil plots by means of the chamber method. Also, selectively, photos of the canopy were made for subsequent determination of the green vegetation fraction. The research sites are characterized in Sect. 2.1; the field management is in Sect. 2.2; the field measurements are in Sect. 2.3; the laboratory measurements is in Sect. 2.4; and the corresponding data file structure is presented in Tables 7–20 in Sect. 3. The location of the research stations and plots followed practical considerations.

### 2.1 Site description

Measurements were performed in two research areas in two study regions, Kraichgau (48.9° N, 8.7° E; 319 m a.s.l.) and

https://doi.org/10.5194/essd-13-1-2021

Earth Syst. Sci. Data, 13, 1–30, 2021

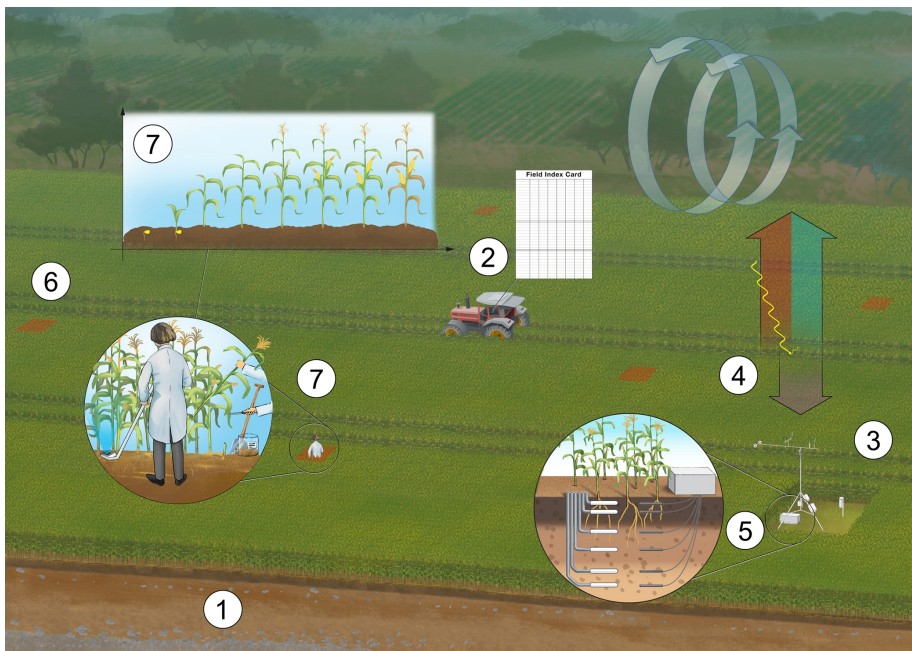

**Figure 2.** Schema of the measurement campaign at the research sites: 1 – soil profile characteristics; 2 – management and cultivation data (sowing date, harvest date, crop type and variety, fertilization, and pesticide application including amount and type) and soil tillage; 3 – meteorological data (rain and air temperature at 2 m height and relative humidity); 4 – soil–/biosphere–atmosphere fluxes using fully equipped eddy-covariance stations for carbon, energy and water vapour flux measurements, as well as wind speed and wind direction; 5 – soil state measurements including water content, temperature and matric potential, soil profile depth permitting at 5, 15, 45, 75, 90 and 130 cm soil depth; 6 – five plots per research site for carbon and nitrogen measurements integrated over depths of 0–30, 30–60 and 60–90 cm; and 7 – plant performance also determined at the plots (phenology, height and leaf area index; yield; aboveground biomass; and carbon and nitrogen in vegetative and generative biomass). A detailed GIS (geographic information system) CE7 data model is included in the dataset including fields, measurements locations and plots. Illustration by TS3 H. Vanselow (http://www.holgervanselow.de/, last access: TS4 ).

the Central Swabian Alb (48.5° N, 9.8° E; 690 m a.s.l.). Each research area comprised three arable fields (in the following research sites) managed by local farmers. In the following, we give a detailed description of the study regions at large and the research sites in particular. The study regions, research areas and study sites are shown in Fig. 3.

### 2.1.1  Kraichgau sites

Kraichgau is a hilly region with fertile soils in the northwest of the state of Baden-Württemberg, southwestern Germany. It is part of the Neckar catchment and borders on the Odenwald low mountains in the north, the Neckar Valley in the northeast, the Stromberg and Heuchelberg downlands in the southeast, the Black Forest in the southwest, and the Rhine Valley in the west. The natural geographic region of Kraichgau is located at an altitude of 100–400 m a.s.l. and covers approximately 1600 km$^2$.

Due to its location in a basin surrounded by low-mountain ranges, Kraichgau is characterized by a mild climate with an annual mean temperature of more than 9 °C, making it one of the warmest regions in Germany. Mean annual precipitation ranges from 720 to 830 mm, with a prevailing southwest-

erly wind direction. The central research area of Katharinenthalerhof contains one ongoing research site. The research area is located close to the city of Pforzheim, southwestern Germany (48.92° N, 8.70° E). The area (385 m a.s.l.) is open and flat, and the prevailing wind direction is southwesterly. The parent soil material genesis is loess with a thickness of several metres. Because of temporally stagnant-water conditions, particularly during spring, a Stagnic Luvisols developed (World Reference Base for Soil Resources, WRB; Michéli et al., 2006). The underlying rock material is shell limestone. The groundwater table is located more than 25 m below the surface. Three eddy-covariance stations (EC1, EC2 and EC3) were installed at adjacent fields with the respective areas of 14.9, 23.6 and 15.8 ha (Fig. 3): EC1 (48°55′42.60″ N, 8°42′10.21″ E) from 16 April 2009 to 17 July 2018, EC2 (48°55′39.99″ N, 8°42′32.03″ E) from 17 April 2009 to 29 October 2018 and EC3 (48°55′38.05″ N, 8°42′57.37″ E) from 8 May 2009 to 17 September 2018, with dates indicating the time span of included measurements. Physicochemical properties of the soils are provided in Table 1. Photos show that EC1 to EC3 were meadows until the 1960s, after which the area at large was drained and agricultural fields were established.

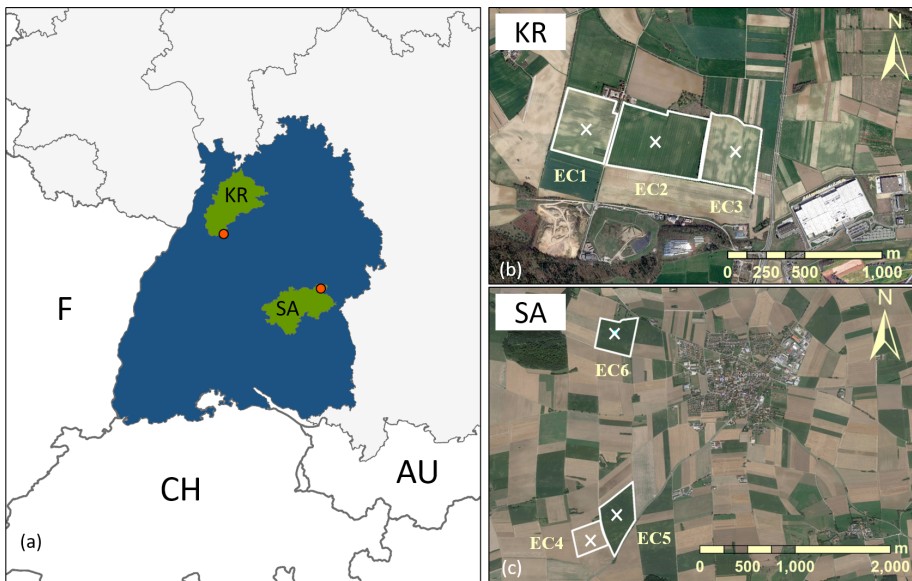

**Figure 3. (a)** Geographical overview and locations of the study sites and EC stations in **(b)** Kraichgau (KR) and **(c)** the Swabian Alb (SA; modified from Eshonkulov et al., 2019; © Google Earth: KR on 31 March 2017 and SA on 26 August 2016).

### 2.1.2 Swabian Alb sites

The low-mountain range of the Swabian Alb is a region with an approximate width of up to 40 km that stretches in a southwest–northeast direction over approximately 220 km, from the Black Forest in the southwest to the Franconian Alb in the northeast, covering an area of ca. 5700 km$^2$ in the state of Baden-Württemberg. To the northwest, the Swabian Alb is separated from the foothills by a 300–400 m high escarpment (Albtrauf). To the southeast, the Danube Valley forms the border to the geographic region of Upper Swabia (Oberschwaben). The Swabian Alb is structured in several geographic regions. Its central part is subdivided into the Mittlere Kuppenalb and the Mittlere Flächenalb. The Mittlere Kuppenalb in the northwest is characterized by mountains forming a hilly plateau which reaches elevations of 800–850 m a.s.l. The Mittlere Flächenalb in the southeastern part has a more levelled relief descending from about 650–750 m a.s.l. to the Danube Valley at about 520 m a.s.l. Due to its altitude, the climate of the Kuppenalb is much colder and harsher than that of the foothills. The mean temperature is 6–7 °C, i.e. lower by about 2 °C than in Kraichgau. Basins of cold air are typical for this cliffy karst region, where night frost may occur even during summer months. Mean annual precipitation, which falls mostly during summer, ranges from 800 to 1000 mm. Prevailing wind direction is westerly to southwesterly. Due to its lower elevation the Flächenalb is slightly warmer and drier.

The Swabian Alb is the largest contiguous karst region in Germany. The foothills are mostly formed by Black Jurassic and the escarpment by Brown Jurassic, whereas the plateau consists of White Jurassic. The in situ unlayered reef limestones (*Massenkalke*) and dolomites are the reason for the deep karstification and the development of the hilly plateau crossed by widely ramified dry valleys. Depending on the bedrock, soils are clayey loams (*tonige Lehme*) and calcareous rendzina (*Kalkscherbenböden*) or shallow calcareous black soils (*Kalkstein-Schwarzerden*), in the dry valleys decalcified loams. The soils are classified as a Calcic Luvisol at research site 4, Anthrosol at research site 5 and Rendzic Leptosol at research site 6 according to the World Reference Base for Soil Resources (WRB; Michéli et al., 2006).

The intensive agricultural land use in this area (Wöhling et al., 2013a) is characterized by a relatively balanced mix of crop production, dairy farming, bull fattening, pig production and biogas production. Most farm holdings simultaneously produce three to five different crops, with spring barley, winter wheat, winter barley and winter rapeseed being the dominant crops, while dairy and cattle farmers tend to also grow silage maize, clover and field grass (Troost and Berger, 2015). Three EC stations (EC4, EC5 and EC6) were installed at research sites 4–6 with respective areas of 8.7, 16.7 and 13.4 ha (Fig. 2): EC4 (48°31′38.95″ N, 9°46′9.73″ E; 685 m a.s.l.) from 30 April 2009 to 31 August 2018, EC5 (48°31′47.50″ N, 9°46′23.41″ E; 687 m a.s.l) from 30 April 2009 to 2 August 2018 and EC6 (48°32′49.29″ N, 9°46′23.16″ E; 692 m a.s.l.) from 30 April 2009 to 26 July 2018. Both EC4 and EC5 have been used as agricultural fields since the 1970s. It is likely that they were subject to land consolidation in the 1980s. Based on personal accounts, EC6 is known to have been under agricultural operation since at least the 1940s. Before the land consolidation in 1987, the field was separated into at least 20 different

plots. The current owner has been using the field at EC6 since 1993. Research sites 2 and 4 are currently still in operation.

## 2.2 Field management and description

Basic field management information was provided by the farmers directly as field card index data. These contained information on crop rotations (Table 2), fertilization, soil management and pesticide usage (Table 3). Crop yield is reported as total generative biomass at harvest by the farmers. In Germany, grain yields are expected to have a residual water content of around 14 %. Separately, vegetative and generative biomass was also determined by plot sampling as part of the biomass characterization (cf. Sect. 2.3.4). From Table 3 it can be seen that the yields reported by the farmers are commonly lower than those reported by the scientists, which were determined at the experimental plots (cf. Sect. 2.3.4). High discrepancies of yields are found in the data of 2018 at EC2, EC4 and EC6, with inexplicably low yields reported for the plot replicates. In 2013, EC4 had a winter rapeseed yield of $1.2\,\mathrm{t\,ha^{-1}}$, which was attributed to hail damage.

Under the assumptions that no silage maize was used as fodder, that the water content of the harvested maize was 70 % by mass, that the amount of carbon in the biogas digestate was 17.4 % of the carbon exported from the field (Lindorfer and Frauz, 2015) and that the organic carbon content was 58 % of maize organic matter, we note the following: harvest and fertilization data provided by the farmers and included in the dataset indicate that between the silage maize carbon exported and returned to the field with the biogas digestate, referenced to the 15 silage maize cultivation periods, on average, approximately $900\,\mathrm{kg\,ha^{-1}}$ carbon were unaccounted for, indicating they may have been used as fertilizers elsewhere. It is worth noting that the cover crops are not accounted for in the carbon balance. Physicochemical properties of the fertilizers are given in Tables A1–A2 TS10 and were used to calculate the input of nitrogen and organic matter to the fields.

Note that the yield values reported in last two columns of Table 3 differ from each other. In the "field" column values are farmer reported yields for the entire field. These are affected by harvest losses, no yields on tractor tracks and reduced yields due to side effects on the field. In the "plot" column the reported values stem from observations on experimental plots located far away from the edge of the field and between tractor tracks. This explains why the farmer values are mostly smaller than the plot values. Values in brackets are standard deviations over the plots. For silage maize in the Kraichgau region, the farmer values are reported as fresh mass.

Crop management was fairly typical for conventional intensive crop production in the areas. Noteworthy is the importance of biogas production, both as a motive for silage maize production and as a supplier of organic fertilizer. Choice of maize and wheat varieties in the sample reflects the climatic differences between the two locations. Due to the shorter growing season, early-maturing silage maize varieties (S220–S240) were preferred at the Swabian Alb sites, while the Kraichgau sites are dominated by medium- to late-maturing varieties (S240–S310). With respect to winter wheat, the full spectrum of varieties ranging from hard (high protein/gluten, German classification group E) to soft (low protein, group C) varieties can be found. Wheat variety choice tends towards the higher-quality end of the spectrum (groups E and A) in Kraichgau and more towards the lower-quality spectrum (groups B and C) in the Swabian Alb locations. Production of marketable quality wheat requires reliably favourable production conditions. Wheat yields are slightly higher ($0.5\,\mathrm{t\,ha^{-1}}$) on average in the Swabian Alb, which may be a consequence of the higher prevalence of the low-protein wheat varieties (groups B and C), which tend to have higher yields (Tables B1–B3). In comparison to district yields presented in Tables B1–B3, the following can be noted: the two spring barley yields reported by farmers are much higher (20 %–40 %) than district averages for the respective years and considerably higher than typical spring barley yields in Germany. Similarly, winter barley yields on Swabian Alb research sites are about 20 % higher than district averages. Silage maize and winter rapeseed yields at the Kraichgau research sites are in line with district averages. Silage maize yields on the Swabian Alb are difficult to compare to district averages, as farmers reported dry-matter yields. Since the Central Swabian Alb lies at the margin of the maize suitability area, silage maize cannot always be harvested at a stage of ideal maturity, and remaining water content cannot be assumed to always have reached literature values (this was not explicitly reported by the farmers). Hence any comparison with wet-matter biomass observations reported at district levels is subject to considerable uncertainty and potential bias.

Table 3 indicates the number of pest and plant control operations between the harvest date of the previous crop and the harvest of the current crop. The number before the colon indicates the number of application; the uppercase letters indicate the type of agent; and the numbers in brackets indicate the number of agents applied. For example, at EC1 in 2010 herbicides were applied once, with three different agents. In the file plant_protection.csv, active substances of agent and application rates are further specified (cf. Table 3).

## 2.3 Field measurements

TS11

## 2.3.1 Meteorological data

Meteorological data were measured at all eddy-covariance stations and recorded on CR3000 data loggers (Campbell Scientific Inc., Logan, UT, USA) in 30 min intervals. Global radiation (Rg) and net radiation were measured with four-

component net radiometers (NR01, Hukseflux Thermal Sensors B.V., Delft, the Netherlands) that were installed about 1.5 m above the canopy. Air temperature and humidity were measured at 2 m height (HMP45, Vaisala Inc., Helsinki, Finland; EC2 from September 2016 and EC1 from December 2016: HC2S3 HygroClip2, Rotronic GmbH, Ettlingen, Germany) and precipitation at 1 m height (ARG100, EML, North Shields, UK). During the period 11 April–2 November 2017, different sensors were used at EC1. During this time, long- and short-wave radiation was measured with a four-component CNR4 net radiometer (Kipp & Zonen B.V., Delft, the Netherlands) and an HMP155 probe (Vaisala Inc., Helsinki, Finland) was used to measure air temperature and humidity. Data were stored on an XLite 9210 data logger (Sutron Corporation, Sterling, VA, USA). The convention used in this dataset is that all energy components directed away from the surface are positive. April–June mean temperature and precipitation sum is presented in Fig. 3, which highlights the mean differences between the two regions. The interannual variability of precipitation is high, whereas that of temperature is low. Mean air June temperatures gradually increased over the reported 10 years. Rain and soil water content as well as soil temperature are shown as an example in Fig. 4. The weather data gap filling and flags were done using an automated Fortran programme, which we summarize here. For all variables no gap filling is marked by flag 0. Gap filling was first tried by using data from an adjacent station. The gap-filled data were then flagged as 11, 12, 13, 14, 15 or 16 for data from EC1 through EC6, respectively. If no wind speed or wind direction data from the adjacent stations were available, a random wind speed was sampled from the data of the previous 12 h (flag = 6). Air temperature was filled by linear interpolation if the data gap was no more than three measurements and correspondingly flagged by 1–3. In other cases, data from neighbouring stations were used. The humidity data were treated in the same way as air temperature, but if a missing value is > 99 % the gap is filled with 99.9 (flag = 5). The air pressure data gaps were also filled like the air temperature. If no data from neighbouring stations are available either, the pressure is set to the average pressure of the region (flag = 7). For the downwelling (down) and upwelling (up) short-wave radiation (rs) and long-wave radiation (rl) the following gap-filling approach was done: for rs_down and rs_up, data gaps are filled with data from neighbouring stations. For rs_up, data points are computed based on the albedo of the previous dataset and rs_down (flag = 4). rl_up was filled in the same way as air temperature. Additionally, rl_up values were checked for plausibility. If values were below 200 W m$^{-2}$, gaps were filled up with data. Precipitation data were gap-filled in the same way as rs_down.

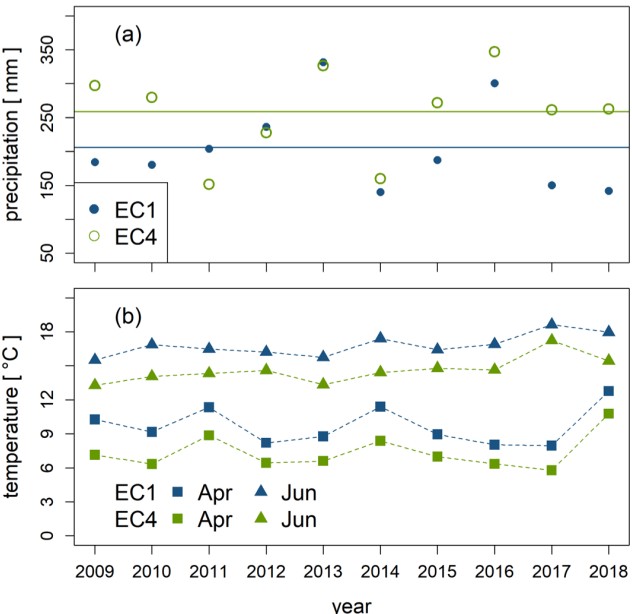

**Figure 4. (a)** Selected meteorological variables at EC1 and EC4. Mean cumulative April–June precipitation; lines indicate the respective means. The years 2014, 2015 and 2018 show very dry growing-season precipitation in KR (EC1) and in 2011 and 2014 in SA CE12 (EC4). The difference in mean precipitation is 60 mm. **(b)** April and June mean air temperatures for 2009 to 2018. SA is 2.2 (.5) TS12 °C cooler than KR. The years 2011, 2014 and 2018 showed very warm April months. The June temperature increases gradually from 2009 to 2018.

### 2.3.2 Surface–atmosphere fluxes

All six EC stations were equipped with the same equipment (Table 4), except for the number of soil sensors which was variable (Table 5). Surface–atmosphere fluxes (net $CO_2$ flux, sensible and latent heat flux) were measured with the eddy-covariance (EC) technique. Each EC station was installed in the centre of a field (Fig. 3) and equipped with an LI-7500 open-path infrared $CO_2/H_2O$ gas analyser (LI-COR Biosciences Inc., USA) and a CSAT3 3D sonic anemometer (Campbell Scientific Inc., UK). The measuring height was adjusted relative to the canopy height. The EC data were logged in 10 Hz resolution on a CR3000 data logger (Campbell Scientific Inc., Logan, UT, USA). The EC data were aggregated to 30 min (raw data available upon request). All other sensor data were stored in 30 min intervals. At EC1, a different system was used during the period 11 April–2 November 2017. During this time, this EC station was equipped with an LI-7200RS enclosed-path $CO_2/H_2O$ gas analyser (LI-COR Biosciences Inc., Lincoln, NE, USA) and an HS-50 3D sonic anemometer (Gill Instruments Ltd., Hampshire, UK). Power supply for all EC stations was ensured by two solar-power batteries with capacities of 12 V and 250 Ah each (Keckeisen Akkumulatoren e.K., Memmingen, Germany). The batteries were charged by four 20 W so-

Please note the remarks at the end of the manuscript.

lar panels (SP20, Campbell Scientific Inc., Logan, UT, USA) at each station. During periods with low solar altitude, however, the power supply was generally insufficient to ensure the operation of the LI-7500. For this reason, direct methanol fuel cell systems with 45 W maximum power supply (EFOY Pro 800 Duo, SFC Energy AG, Brunnthal-Nord, Germany) were installed at EC2 and EC6 in autumn 2015, enabling measurements of the surface–atmosphere fluxes at these locations also in winter.

The EC data from April 2009 to December 2012 were processed using the EC software package TK2 and after January 2013 using version TK3.1 (Mauder and Foken, 2015). Fluxes were computed from 30 min covariances between vertical wind velocity and the corresponding scalar ($CO_2$ concentration, air temperature or humidity). In the TK software, we used the following settings: spike detection (i.e. values exceeding 3.5 times the standard deviation of the last 10 values were labelled as spike); the planar-fit method for coordinate rotation with time periods between 7 to 12 d; Moore (1986) correction except for the longitudinal separation, which was taken into account by maximizing the covariances; the Schotanus et al. (1983) procedure for converting the sonic into actual temperature; and density correction as suggested by Webb et al. (1980). For data quality analysis we used the nine-flag system of Foken (2006). Half-hourly fluxes with flags 7–9 (poor quality data) for friction velocity, sensible heat flux or latent heat flux were excluded. For additional despiking of half-hourly fluxes, we applied a median filter using the median of absolute fluxes of the previous 4 d; fluxes that were > 5 times this median were discarded. For this and for the gap filling, we used the R package REddyProc (Wutzler et al., 2020).

Data gaps occurred in times of sensor excavation for harvest and sowing and due to browsing by animals. The stony ground at the SA sites made it necessary to install soil sensors at EC4 and EC5 at a maximum of 45 cm depth and at EC6 at 15 cm depth. By way of example, time series of $CO_2$ fluxes are presented in Fig. 6 for EC1, which also presents the data coverage. Data for EC2 to EC6 can be found in the Supplement.

At selected sites some additional measurement campaigns to determine soil surface $CO_2$ flux with chamber measurements was done on both bare fallow and vegetated plots. Both types of soil respiration plots (bare and vegetated) were located close to the plots used for plant observations/biomass harvests. Additional bare-fallow plots were established in 2009, 2010 and 2012 (b09, b10 and b12 CE14, respectively) in the research fields. Plots were kept clear of vegetation during the experiment by manual weeding and periodic spot spraying of glyphosate (Monsanto Agrar, GmbH, Germany). The plots were tilled by hand in a way as to mimic mechanized tillage. In addition to plant residues, the vegetated plots received manure/slurry and mineral fertilizer as organic inputs from the farmers' field management. Since bare plots were maintained through multiple years of the experiment,

the plots were located at the periphery of the fields to be outside of the EC footprint. Each of the three plots was allocated randomly within a third of the field outside the main EC footprint avoiding field edges, tractor pathways and other non-representative spots. Soil surface $CO_2$ flux was measured via portable infrared gas analysers (EGM-2 and EGM-4, PP Systems, Amesbury, MA, USA) with an attached soil temperature probe and soil respiration chamber (Demyan et al., 2016; Laub et al., 2021). The soil respiration flux chamber was 10 cm in diameter with an internal volume of 1171 $cm^{-3}$. Fluxes were measured during the growing period of different years (2009, ($n = 5$); 2010, ($n > 15$); 2014, ($n > 3$); 2015, ($n = 7$); and an intensive approximately weekly campaign May 2012 to June 2013, ($n = 40$)). Six replicate measurements were taken within each subplot (vegetated and bare fallow) during each measurement day. Measurement order of the plots was randomized each day to avoid time-of-day effects. Individual measurements which were greater than 6 times the yearly median value were removed as outliers.

### 2.3.3 Soil sampling, soil heat fluxes, and soil temperature and matric-potential measurements

Adjacent to the EC stations but in the tilled soil, temperature sensors (model 107, Campbell Scientific Inc., UK) were installed at 2, 6, 15, 30 and 45 cm soil depth. To measure the volumetric soil water content and soil matric potential, we installed FDR (frequency domain reflectometry) probes (CS616, Campbell Scientific Inc., UK) and matric-potential sensors (model 253, Campbell Scientific Inc., UK) at 5, 15, 30, 45 and 75 cm depth and selective extra depths at some locations. Three soil heat flux plates (HFP01, Hukseflux Thermal Sensors, the Netherlands) were installed 8 cm below the ground surface. At EC1, self-calibrating heat flux plates (HFP01SC, Campbell Scientific Inc., Logan, UT, USA) at 8 cm depth and HydraProbe II sensors (Stevens Water Monitoring Systems Inc., Portland, OR, USA) for soil volumetric water content and soil temperature at 5, 10 and 15 cm depth were used during the period 11 April–2 November 2017. Soil water content and temperature at 5 cm depth and precipitation are presented in Fig. 4 for EC1 in 2010, where the strong drop in soil water contents around DOY 75 (day of the year) and DOY 350 are both attributed to soil freezing. We did not exclude these data from our dataset intentionally. At EC1 to EC3, the soil water content sensors were calibrated to in situ gravimetric soil water content data. In EC4 to EC6, only the factory-calibrated time series are provided. The remaining sites and years are presented in the Supplement.

To determine total and mineral nitrogen ($NH_4^+$ and $NO_3^-$) and organic carbon, soil samples were taken at least in spring (March–April) and autumn (October–November) from the depths 0–30, 30–60 and 60–90 cm at all six research sites during almost all years. Two replicate samples were taken at five permanent locations at every site. In November 2017, soil bulk density was determined at the six research sites

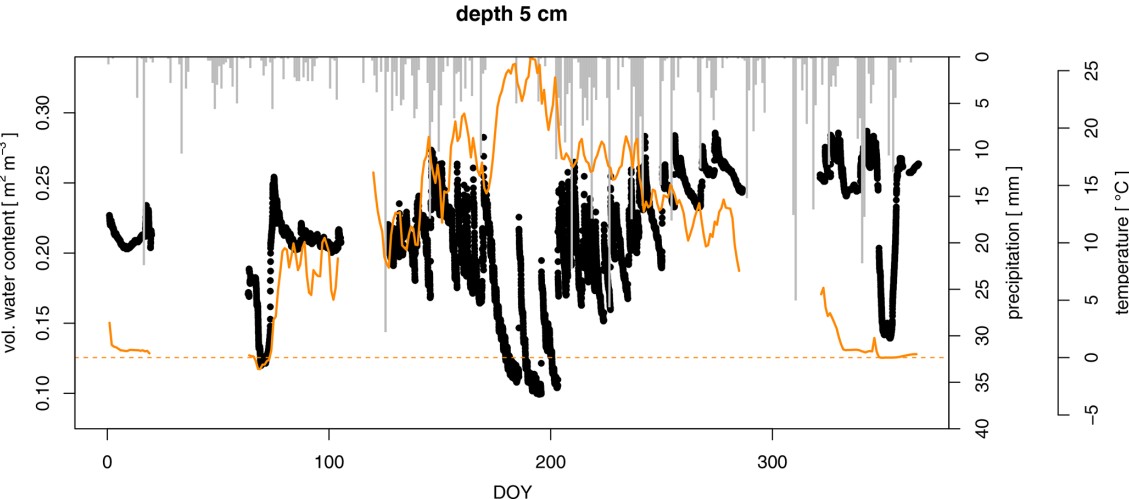

**Figure 5.** Soil water content (black dots), precipitation (grey bars) and soil temperature (orange line) at research site EC1 in 2010 at 5 cm soil depth. The dashed orange line indicates 0 °C. TS13 TS14

(EC1 to EC6) using cylinders with 4 cm height and 100 cm$^3$ volume. Undisturbed cores were taken from depths 0–4, 13–17, 26–30, 36–40 and 46–50 cm if this was possible, depending on soil thickness and stone content. Prior to gravimetric measurement of the water content, soil cores were stored air-tight. For the determination of dry weight, samples were dried at 105 °C until they reached a constant weight. From this we calculated gravimetric water content and bulk density. Using knowledge about bulk density, we CE15 determined bulk density. Samples on bare plots (preparation and maintenance of bare plots is described in Sect. 2.3.2) were collected from 0–30 and 30–60 cm in bare fallow. Composite soil samples were prepared by mixing five soil cores taken within a specific area in each plot into one homogenous sample. The same analysis was performed on the samples from the bare-soil plots as from the vegetated plots.

### 2.3.4   Plant sampling and development variables

At each field, five plots of 4 m$^2$ were randomly selected and permanently marked to track phenological stage and reported on the BBCH CE16 scale (Meier, 2018), total leaf area index, plant height and plant biomass. Phenological stages were assessed at least in 4-weekly intervals during winter and biweekly during the main growth period starting in autumn (winter crops) or early spring (spring crops) until maturity. In each plot of the research fields, observations of phenology and plant height on 10 different plants were made and are reported as plot replicates in the files. In 2017 and 2018, these measurements were carried out less frequently. During the main growth period starting in early spring, total leaf area index and plant height were determined about biweekly until crop maturity at the central square metre of every plot. An LAI-2000 plant canopy analyser (leaf area index; LI-COR

Biosciences Inc., USA) was used to measure total leaf area index. Intermediate harvests of total aboveground biomass took place at stem elongation (decimal code (DC) 31 on the BBCH scale) and full flowering (DC 65) using five plants per plot. At crop harvest (maturity), the biomass of the central square metre of each subplot was cut at ground level and separated into vegetative and generative fractions. From 2017 on, also intermediate harvests were conducted on an areal basis. Biomass was sampled from 0.5 × 0.5 m$^2$ subplots for all crops except for maize, where plants were sampled from 1.5 m sections of the seeded rows and multiplied by row spaces for areal extrapolation. Generative biomass for winter wheat, spelt and barley is only the grain; for the maize it is the grains without the spindle; and for winter it is rapeseed (only the seeds without the pods). The remaining parts of the plants are considered aboveground biomass.

### 2.3.5   Vegetation photos

For ground truth, green vegetation fraction (GVF) was determined based on photos at EC1, EC2 and EC3 fields in 2012 and 2013 (Imukova et al., 2015). The photos are available as part of the dataset. Within each study field, five plots (1 × 1 m$^2$) were permanently marked. During the growing season (April–October), canopy photos were taken from these plots in a weekly resolution. For the photos, a Nikon COOLPIX P7000 digital camera (Nikon Corporation, Tokyo, Japan) was used. Photos were taken from 1 m above the canopy (nadir sampling) using the automatic mode of the camera, with and without flash. This measurement campaign was extended by using RapidEye satellite images to derive high-resolution gridded GVF data (cell grid size of 5 × 5 m$^2$). Satellite images had been provided by the German Aerospace Center (DLR) as a part of the RapidEye Science Archive

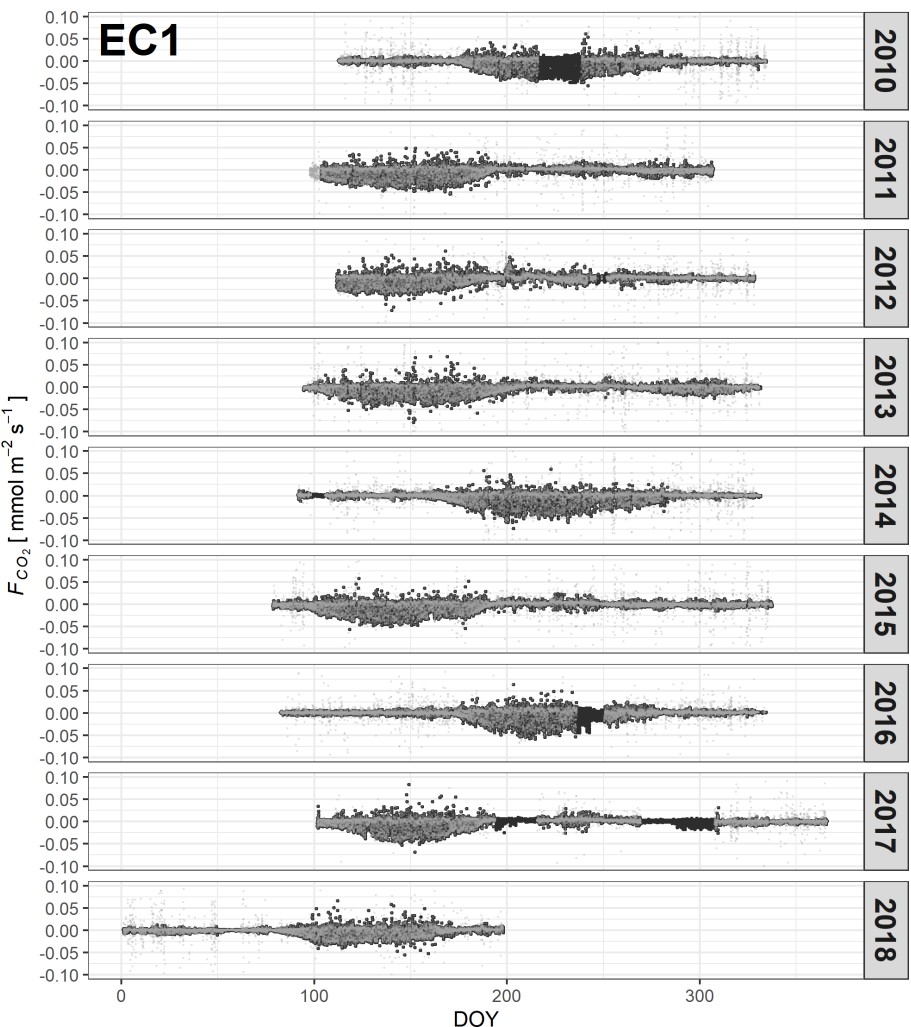

**Figure 6.** $CO_2$ fluxes measured by the eddy-covariance method at EC1 for 2010–2018. Unfiltered measurements are depicted in grey, and gap-filled data are in black.

(proposal 505). Obtained high-resolution GVF grid data revealed a bimodal distribution of GVF at the regional scale during the growing seasons (Imukova et al., 2015). This bimodal behaviour was explained by phenological differences between early-covering (ECC; e.g. winter wheat and winter rapeseed) and late-covering crops (LCC; e.g. maize and sugar beet). The data and derived results imply splitting the generic cropland class in land surface models MP into ECC and LCC as a potential to improve the simulation of energy and water fluxes at the land surface, particularly during the second part of the growing season (Bohm et al., 2020 TS15).

## 2.4   Laboratory measurements

### 2.4.1   Soil organic carbon and nitrogen

Soil microbial biomass C and N ($C_{mic}$ and $N_{mic}$) were analysed on wet samples using the chloroform fumigation extraction (CFE) method (Joergensen, 1996). Briefly, ap-

proximately 20 g non-sieved soil samples were fumigated with ethanol-free chloroform for 24 h. Fumigated and non-fumigated samples were extracted with a 0.5 M $K_2SO_4$ solution. $C_{mic}$ and $N_{mic}$ in extracts were analysed with a multi N / C analyser (Analytik Jena AG, Jena, Germany). Mineral N (ammonium and nitrate) was measured in the non-fumigated 0.5 M $K_2SO_4$ extracts using a flow-injection analyser (FIAstar 5000, FOSS, Denmark). $C_{mic}$ and $N_{mic}$ were calculated from the difference in C and N contents of fumigated and non-fumigated samples using a $k_{EC}$ value of 0.45 and a $k_{EN}$ value of 0.54 (Joergensen, 1996) after correcting for gravimetric moisture content. Total organic C (TOC) in bulk soil was analysed by dry combustion according to DIN ISO 13878 (1998) TS16 with a vario EL III elemental analyser (Elementar, Hanau, Germany). The soil samples from vegetated plots were analysed for organic carbon and total nitrogen content (vario MACRO cube, Elementar Analysensysteme GmbH) as well as for ammonium and nitrate con-

*Earth Syst. Sci. Data*, 13, 1–30, 2021

https://doi.org/10.5194/essd-13-1-2021

centrations. Ammonium was determined photometrically in a nitroprusside–salicylate solution. Nitrate was measured by ion chromatography (861 Advanced Compact IC, Metrohm) equipped with an anion separation column (Metrosep A Supp 5, Metrohm). Gravimetric moisture content was determined by drying approximately 0.1 kg of wet soil at 105 °C for 24 h.

### 2.4.2 Plant biomass, carbon and nitrogen content

Plant material was separated into vegetative and generative fractions. Vegetative parts were dried to a constant weight at 60 °C, and generative parts were dried at 28 °C. After determining dry weights, the generative parts were manually threshed to determine crop yield, whereas the vegetative parts were cut using a chaff cutter and homogeneously mixed. Randomly picked material of vegetative parts and of harvestable products was milled using a laboratory mixer mill (MM 301, Retsch, Haan, Germany). Fine powder of vegetative parts and of harvestable products were analysed for carbon and nitrogen using an elemental analyser (vario EL, Elementar Analysensysteme, Hanau, Germany) as described in Högy et al. (2009). In 2017 and 2018 the residual water content was determined, too. This was achieved by further oven-drying of the samples at 105 °C until a constant weight had been reached.

### 2.5 Measurement uncertainty

In environmental sciences, observations are afflicted with random and systematic errors and additionally by uncertainty due to spatial heterogeneity of the system of interest. In principle, errors and uncertainties can be approximated quantitatively by theoretical and practical approaches. Identifying which part of a measured value has to be attributed to the random error, systematic error or uncertainty can prove highly challenging and is scale dependent. One common approach is by replicating the measurement process. For the weather and eddy-covariance data, details were already given in Sect. 2.3.1 and 2.3.2., respectively. Quality flags in both datasets are qualitative indicators, and for the weather data, only the instrumental measurement uncertainty is known (Table 4), since replicate measurements were not made. Generally, uncertainties in determined height, direction and orientation of measurement devices as well as installation depths of sensors are unresolved. In the predominant cases for the soil and plant measurements, including the soil chamber flux measurements, errors and uncertainties can be deduced from (a) replicate measurements/sampling within a plot or (b) replicate plots in a field. In most cases, replicate measurements are directly provided in the data files; exceptions are the soil profile characterization and the leaf area measurements where the replicate measurements were averaged and their standard deviations are reported. No replicate measurements exist for the time series of soil water content, soil temperature and matric potentials. An exception is the measured

ground heat fluxes which were determined in replicates of three at each station. For some of the analytical instruments, the uncertainties as determined by the manufacturer are given in Table 4. The uncertainties of the remaining measurement devices are not explicitly covered, as they are considered negligible or implicitly covered in replicate measurements. In most cases, systematic errors are sometimes even impossible to quantify. We consider the random measurement error to be captured by the replicate measurements/samples from within a plot, while those between plot replicates are an indicator for effects of heterogeneity. For obvious reasons, over 10 years, different persons were involved in sampling, installing sensors and handling the experiments, sometimes within a season. While the methods remained the same, this has the potential to induce systematic errors, which are not further resolved, since the information is no longer retrievable.

## 3 Scope and structure of the dataset

We provide figures and tables alluding to the scope and nature of the datasets for download at https://doi.org/10.20387/bonares-a0qc-46jc (Weber et al., 2021). The dataset variable structure follows the description in *Error! Reference source not found.* TS17 (flat structure in the BonaRes dataset download). The intent is to provide an overview of the quantified system variables and properties, without needless repetition of previously peer-reviewed analyses. In view of the fact that the dataset contains 17 sub-datasets, the structure is presented here explicitly. All data files were ensured to be machine-readable using the software R and the library data.table and function fread. The column names, units, data types and descriptors are listed in Tables 6–19 TS18 (for the review process, Tables 6–18 TS19 are found in Sect. 12). Numerous figures with time series of the flux and soil measurements are presented in the Supplement. Lastly, a GIS data model comprised of four files identifies the geolocation of the measurements. The research fields are given in 01_research_sites.gpkg; the research stations EC1 to EC6 are in 02_research_stations.gpkg; the research plots of the soil C / N CE17 and plant development measurements are in 03_research_plots.gpkg; and the additional plots for the flux measurements of the chamber measurements and additional soil C / N measurements are in 04_research_plots_chambers.gpkg.

## 4 Data availability

The digital database is available freely for download from the BonaRes Data Centre https://doi.org/10.20387/bonares-a0qc-46jc (Weber et al., 2021).

```
Structure of the dataset

    ./00_metadata           |  information on the data sets

    ./01_ec_flux            |    eddy covariance station data

    ./02_chamber_fluxes     |    weather data (model drivers)

    ./03_weather            |    water content

    ./04_soil               |    soil observations

    ./05_management         |    model drivers, e.g. farm management
        /cultivation        |      sowing, harvest and info on Wheat varieties
        /fertilization      |      C and N-input
        /soil_management    |      model input

    ./06_plant              |    observations of plant data
        /biomass            |      time series of biomass of plants
        /cn                 |      C/N ratios of plant samples
        /lai                |      leaf area index
        /phenology          |      phenological development

    ./07_soil_properties    |    observation of soil data
        /cn                 |      time series of CN measurements
        /profile            |      soil profile description

    ./08_GVF                |  empty, please refer to separate data publication
```

**Figure 7.** Folder structure of the dataset. TS20

## 5   Summary and conclusion

We provide a comprehensive dataset on agricultural crop growth and land surface exchange on arable soils in Germany. The continuous eddy-covariance measurements on adjacent fields and the long duration of our measurements (2009–2018) is unique and allows for new insights into the role of crop rotations for land surface exchange processes. According to a recent report by the Alliance of Science Organisations in Germany our installations have been the only ones on agricultural land throughout southern Germany that fulfil the criteria for becoming part of the intended national observatory network for terrestrial ecosystem research (Kögel-Knabner et al., 2018). One research site per region (EC2 and EC4) is still fully operational, while the remaining sites were dismantled after completion of the project at the end of the growing season in 2018.

We recognized that the interannual variability within locations exceeds the effect of regional climate. In other words, a direct comparison of fluxes measured in the two study regions is only possible if the measurements are performed in the same year under comparable large-scale weather conditions (Wizemann et al., 2015). Although some drier growing seasons were identified with sometimes low soil volumetric water contents in the upper soil layers, it became apparent that the deep loess soil profiles in the Kraichgau region and the soils in the cooler and wetter Swabian Alb region were generally not severely water-deficient. An exception to that was the very early ripening and subsequently harvest of maize in the Kraichgau region in 2018.

The dataset was used extensively to calibrate soil–crop models and land surface models. In spite of the high data quality and the extensive coverage of crops and years, we would like to draw the attention to some possible improvements for future campaigns like the one presented. First, it became apparent that it would be beneficial to include measurements to infer information on the partitioning between the evaporation and transpiration of the crops. Also, we notice that, due to a solar-power shortage in winter, we have some data gaps in the EC measurements. We think it would be worthwhile to extend the research by extending the measurements on soil (hydraulic) properties (transience, hydrophobicity, structure, etc.). In the future, it would be beneficial to properly quantify the contribution of the cover crops to the overall fluxes and budgets, as well as to include sensors that capture the $N_2O$ emission. Extending the monitoring of root growth as well as of root water uptake and root decay by more detailed, continuous measurements would very valuable for model improvement. We also recommend regularly including measurements on the water content of the crops for remote sensing applications. Finally, we note that the measurements will be continued at two research sites, EC2 and EC4, that is, at one site in each region.

**Table 1.** Soil characteristics at the six research sites EC1 to EC6 (data are presented in profile_data.csv, and methods are described in Sect. 2.3.3).

| EC | Ht cm | Hl cm | Hor CE8 | bd g cm$^{-3}$ | por − | fc − | wp − | stc − | s wt% CE9 | u wt% | t wt% | class | som wt% | lc wt% | pH |
|---|---|---|---|---|---|---|---|---|---|---|---|---|---|---|---|
| 1 | 0 | 32 | Ap | 1.37 | 0.483 | 0.369 | 0.162 | < 1 | 2.5 | 79.4 | 18.1 | Ut4 | 1.75 | 1.5 | 6.9 |
|  | 32 | 48 | Sw-M | 1.51 | 0.43 | 0.365 | 0.223 | < 1 | 2.0 | 79.2 | 18.8 | Ut4 | 0.61 | 0.43 | 6.7 |
|  | 48 | > 90 | M-Sw | 1.48 | 0.442 | 0.404 | 0.243 | < 1 | 0.9 | 80.4 | 18.7 | Ut4 | 0.42 | 0.34 | 6.6 |
| 2 | 0 | 33 | Ap | 1.33 | 0.498 | 0.343 | 0.153 | < 1 | 2.6 | 79.5 | 17.9 | Ut4 | 1.53 | 1.38 | 6.2 |
|  | 33 | 72 | Sw-M | 1.46 | 0.449 | 0.371 | 0.223 | < 1 | 2.9 | 77 | 20.1 | Ut4 | 0.52 | 0.43 | 6.4 |
|  | 72 | > 90 | M-Sw | 1.53 | 0.423 | 0.417 | 0.239 | < 1 | 1.6 | 79.7 | 18.7 | Ut4 | 0.34 | 0.28 | 6.5 |
| 3 | 0 | 30 | Ap | 1.37 | 0.483 | 0.338 | 0.159 | < 1 | 1.8 | 81.1 | 17.1 | Ut4 | 1.64 | 1.46 | 6.4 |
|  | 31 | 60 | Sw-M | 1.5 | 0.434 | 0.355 | 0.163 | < 1 | 1.0 | 80.4 | 18.6 | Ut4 | 0.83 | 0.59 | 6.5 |
|  | 60 | > 90 | M-Sw | 1.51 | 0.43 | 0.358 | 0.137 | < 1 | 0.8 | 83 | 16.1 | Ut3 | 0.63 | 0.45 | 6.6 |
| 4 | 0 | 21 | Ap1 | 1.31 | 0.506 | 0.408 | 0.217 | 1_2 | 6.2 | 56 | 37.8 | Tu3 | 4.35 | 3.2 | 6.9 |
|  | 21 | 29 | Ap2 | 1.34 | 0.494 | 0.37 | 0.331 | 1_2 | 8.9 | 52.5 | 38.6 | Tu3 | 2.13 | 2.04 | 6.8 |
|  | 29 | 41 | Tv | 1.32 | 0.502 | 0.395 | 0.304 | 1_2 | 8.4 | 43.3 | 48.4 | Tu2 | 1.63 | 1.37 | 6.7 |
|  | 41 | NA | cxC | NA | NA | NA | NA | > 50 | NA | NA | NA | NA | NA | NA | NA |
| 5 | 0 | 20 | Ap | 1.37 | 0.483 | 0.403 | 0.2 | < 1 | 2.8 | 68.3 | 28.9 | Tu4 | 3.64 | 2.95 | 6.4 |
|  | 20 | 60 | M1 | 1.4 | 0.472 | 0.335 | 0.21 | < 1 | 2.1 | 64.3 | 33.6 | Tu3 | 1.44 | 1.4 | 6.4 |
|  | 60 | 90 | M2 | 1.51 | 0.43 | 0.417 | 0.302 | < 1 | 1.8 | 64 | 34.2 | Tu3 | 0.71 | 0.56 | 6.2 |
| 6 | 0 | 12 | Ap1 | 1.04 | 0.608 | 0.384 | 0.228 | 2_5 | 3.2 | 51.2 | 45.6 | Tu2 | 5.5 | 5.57 | 6.9 |
|  | 12 | 21 | Ap2 | 1.29 | 0.513 | 0.422 | 0.228 | 5_10 | 4.1 | 48.3 | 47.6 | Tu2 | 3.88 | 3.87 | 7.1 |
|  | 21 | NA | NA | NA | NA | NA | NA | NA | NA | NA | NA | NA | NA | NA | NA |

EC: eddy-covariance station, i.e. research site. Ht: top depth of soil horizon. Hl: lower depth of soil horizon. bd: bulk density. por: porosity. fc: field capacity. wp: wilting point. stc: stone content. S: sand. u: silts. t: clay. class: soil texture class (soil organic matter at the beginning of the research period in 2009). lc: lime content. wt %: weight percentage. NA: not available. TS5 The soil texture and texture classes are in reference to the German soil classification (Sponagel, 2005). Model parameters for the van Genuchten–Mualem (van Genuchten, 1980 TS6) soil hydraulic functions can be derived with a new pedotransfer function (Szabó et al., 2021), which includes uncertainties. Description of soil hydraulic properties over the full moisture range can be achieved using the Brunswick model (Weber et al., 2019; Streck and Weber, 2020) in conjunction with the pedotransfer function by Weber et al. (2020).

**Table 2.** Research sites EC1 to EC6 and land use from 2010 to 2018.

| Year of harvest | Kraichgau | | | Swabian Alb | | | Crop | Frequency |
|---|---|---|---|---|---|---|---|---|
|  | EC1 14.9 ha | EC2 23.6 ha | EC3 15.8 ha | EC4 8.7 ha | EC5 16.7 ha | EC6 13.4 ha | | |
| 2010 | SM | WR | WW | WR | WW | CC-SM | CC-SM | 13 |
| 2011 | WW | WW | CC-SM | WW | CC-SM | WW | SM | 2 |
| 2012 | WR | CC-SM | WW | CC-SB | SM | WB | CC-GM | 1 |
| 2013 | WW | WW | WR | WR | WB | CC-SM | WW | 22 |
| 2014 | CC-SM | CC-SM | WW | WW | SP | WW | SP | 1 |
| 2015 | WW | WW | CC-SM | WW | CC-SM | WB | WB | 5 |
| 2016 | CC-GM | WR | WW | CC-SB | CC-SM | CC-SM | CC-SB | 2 |
| 2017 | WW | WW | WW | CC-SM | WB | WW | WR | 8 |
| 2018 | WR | WW | CC-SM | WW | WR | WB | Sum | 54 |

SM: silage maize. GM: grain maize. WR: winter rapeseed. WW: winter wheat. SP: spelt. CC: cover crop. WB: winter barley. SB: spring barley.

**Table 3.** Summary of field management and nitrogen and organic matter input.

| Site | Year | Crop[a] | | | Fertilization Total N[b] and OM[c] input | | Pest and plant control[d] | Yield[e] Field | Plot |
|------|------|------|------|------|------|------|------|------|------|
| | | Code | Cultivar | Group | kg N ha⁻¹ | kg C ha⁻¹ | Times and type | Mg ha⁻¹ | Mg ha⁻¹ |
| EC1 | 2010 | SM | Cannavaro | S310 | 29.2 (0) | 0 (0) | 1: H(3) | 42.0 | 19.83 (2.38) |
| | 2011 | WW | Akteur | E | 169.8 (169.8) | 0 (0) | 3: F(2), H(2) | 8.41 | 10.25 (0.55) |
| | 2012 | WR | Elado_Artoga CE10 | – | 210.5 (210.5) | 0 (0) | 3: F(1), H(3) | 4.61 | 4.78 (0.44) |
| | 2013 | WW | Akteur | E | 179.2 (179.2) | 0 (0) | 5: F(3), G(1), H(2) | 8.64 | 8.86 (0.94) |
| | 2014 | CC–SM | Grosso | S250 | 51.2 (283) | 0 (1767) | 1: H(3) | 52.6 | 23.24 (1.75) |
| | 2015 | WW | Sokal | A | 265.8 (265.8) | 349 (894) | 3: F(3), G(1), H(3) | 8.5 | 9.44 (1.58) |
| | 2016 | CC–GM | NAN | – | 0 (219) | 0 (0) | 1: H(2) | 11.0 | 14.06 (1.62) |
| | 2017 | WW | Patras | A | 186 (186) | 0 (0) | 3: F(4), G(2), H(2), I(1) | 7.8 | 8.21 (0.87) |
| | 2018 | WR | Alicante, Graf | – | 373.4 (373.4) | 968 (2482) | 4: F(1), H(3), I(1) | 4.2 | 4.31 (0.69) |
| EC2 | 2010 | WR | NK Flair | – | 268.1 (268.1) | 0 (0) | 1: F(1), I(1) | 3.853 | 4.45 (0.2) |
| | 2011 | WW | Akteur | E | 169.8 (169.8) | 0 (0) | 3: F(3), H(2) | 8.95 | 9.44 (0.23) |
| | 2012 | CC–SM | Cannavaro | S310 | 109 (264) | 0 (975) | 1: H(3) | 56.5 | 24.5 (1.4) |
| | 2013 | WW | Akteur | E | 177.8 (177.8) | 0 (0) | 4: F(3), H(2) | 7.65 | 7.91 (0.39) |
| | 2014 | CC–SM | Grosso | S250 | 210 (210) | 0 (0) | 1: H(3) | 49.5 | 23.1 (1.25) |
| | 2015 | WW | Akteur | E | 260.4 (260.4) | 274 (702) | 3: F(3), G(1), H(3) | 8.7 | 10.15 (0.88) |
| | 2016 | WR | PR 46 W 26 | – | 173 (173) | 0 (0) | 3: F(1), H(4), I(1) | 3.2 | 4.37 (0.97) |
| | 2017 | WW | Sokal | A | 186 (186) | 0 (0) | 2: F(3), G(1), I(1) | 9.2 | 9.5 (1.7) |
| | 2018 | WW | Patras, Sokal | A, A | 185.7 (185.7) | 0 (0) | 3: F(2), G(2), H(2), I(1) | 9.1 | 2.6 (0.55)[g] TS7 |
| EC3 | 2010 | WW | Cubus | A | 220.1 (220.1) | 0 (0) | 2: F(1), H(2), I(1) | 7.11 | 7.95 (0.92) |
| | 2011 | CC–SM | Cannavaro | S310 | 40.5 (204) | 0 (0) | 1: H(3) | 58.5 | 25.51 (1.85) |
| | 2012 | WW | Akteur | E | 203.5 (203.5) | 0 (0) | 3: F(3), G(1), H(2), I(1) | 7.82 | 9.94 (0.77) |
| | 2013 | WR | Alabaster, Fregat | – | 235.1 (235.1) | 0 (0) | 3: F(1), H(2), I(1) | 4.39 | 7.21 (0.58) |
| | 2014 | WW | JB Asano | A | 198.2 (198.2) | 0 (0) | 2: F(2), H(1) | 8.34 | 10.47 (1.15) |
| | 2015 | CC–SM | P 8509 | – | 138 (203) | 0 (0) | 2: H(3) | 46.8 | 21.94 (2.15) |
| | 2016 | WW | Estivus Pamier Ferrum | A, A, B | 206 (206) | 0 (0) | 2: F(3), G(3), H(2) | 7.7 | 6.88 (0.93) |
| | 2017 | WW | Estivus | A | 186 (186) | 0 (0) | 3: F(2), G(2), H(1), I(1) | 6.0 | 6.21 (1.47) |
| | 2018 | CC-SM | Various | | 0 (202) | 0 (438) | 1: H(3) | 53.0 | NA[h] |
| EC4 | 2010 | WR | Visby | | 210.9 (210.9) | 0 (0) | 6: F(4), G(2), H(4), I(3) | 2.9 | 2.82 (0.53) |
| | 2011 | WW | Akteur | E | 253 (253) | 0 (0) | 4: F(2), F(4), H(5), I(1) | 8 | 8.33 (1.94) |
| | 2012 | CC–SB | Summer | | 65.5 (93.6) | 0 (0) | 3: F(3), H(3) | 8.5 | 8.86 (0.36) |
| | 2013 | WR | PR 49 W 20 | | 115.5 (133.5) | 0 (0) | 3: F(2), I(3) | 1.2 | 1.83 (1.06) |
| | 2014 | WW | Orcas | B | 202 (202) | 0 (0) | 2: G(2), H(3) | 8.5 | 10.42 (0.41) |
| | 2015 | WW | Arezzo | B | 226.5 (226.5) | 0 (0) | 3: F(3), G(1), H(1), I(1) | 9.0 | 9.93 (0.97) |
| | 2016 | CC–SB | Grace | | 122.2 (289.4) | 0 (0) | 3: F(2), H(4) | 8.1 | 6.22 (0.75) |
| | 2017 | CC–SM | LG 30 238 | S220 | 114 (157) | 3412 (1314) | 2: H(2) | 15[i] | 30.6 (13.24) |
| | 2018 | WW | Porthus | B | 195.7 (195.7) | 341.6 (876) | 3: F(4), G(2), H(3) | 10.4 | 3.15 (0.27)[g] |
| EC5 | 2010 | WW | Pamier | A | 253 (253) | 152.1 (390) | 3: F(3), G(1), H(4) | 7.9 | 9.05 (1.16) |
| | 2011 | CC–SM | Agro-Yoko | S240 | 206 (206) | 138.5 (355) | 1: H(2) | 21 | 18.66 (1.6) |
| | 2012 | SM | Amanatidis | S220 | 97.4 (267.4) | 0 (1065) | 1: H(2) | 17.2[i] | 14.46 (1.72) |
| | 2013 | WB | Hobbit | | 270 (269.5) | 249.2 (639) | 3: F(3), G(1), H(2) | 9.7 | 8.75 (0.29) |
| | 2014 | SP | Frankenkorn | | 170 (170) | 276.9 (710) | 2: F(1), G(1), H(2) | 9.0 | 6.58 (1.12) |
| | 2015 | CC–SM | LG 30.217 | S220 | 70.2 (162.2) | 0 (0) | 2: H(3) | 16.2[i] | 17.26 (2.64) |
| | 2016 | CC–SM | LG 30.217 | S220 | 22.5 (151.3) | 0 (0) | 1: H(1) | 17.4[i] | 21.85 (1.52) |
| | 2017 | WB | Wotan | – | 183 (183) | 305 (781) | 3: F(1), H(3) | 8.9 | 10.46 (2.36) |
| | 2018 | WR | Bender | – | 208 (308.4) | 277 (1349) | 2: H(2) | 4.7 | 5.37 (1.03) |

https://doi.org/10.5194/essd-13-1-2021

| Site | Year | Crop[a] | | | Fertilization Total N[b] and OM[c] input | | Pest and plant control[d] | Yield[e] Field | Plot |
|------|------|------|------|------|------|------|------|------|------|
| | | Code | Cultivar | Group | kg N ha$^{-1}$ | kg C ha$^{-1}$ | Times and type | Mg ha$^{-1}$ | Mg ha$^{-1}$ |
| EC6 | 2010 | CC-SM | Fernandez, PR 39 A 98 | S250, S240 | 90 (219) | 0 (1110) | 1: H(3) | 13.8[i] | 14.77 (3.07) |
| | 2011 | WW | Hermann | C | 220.8 (220.8) | 0 (0) | 3: F(3), H(1) | 8.88 | 9.49 (0.93) |
| | 2012 | WB | Winter | | 281.2 (281.2) | 577.2 (1480) | 2: F(3), G(1) | 8.8 | 8.61 (1.29) |
| | 2013 | CC–SM | SY Kairo, Agro-Yoko | S240, S210 | 127.5 (261) | 585 (2055) | 3: H(6) | NA | 13.8 (2.32) |
| | 2014 | WW | Pamier | A | 229.5 (229.5) | 0 (0) | 3: F(3), G(1), H(1) | 9.9 | 9.3 (2.1) |
| | 2015 | WB | Anisette | – | 198.2 (198.2) | 360.8 (925) | 4: F(3), G(1), H(3) | 7.2 | 8.37 (1.31) |
| | 2016 | CC–SM | Toninio | S230 | 151 (282.8) | 398 (1576) | 3: H(5) | 18.0[i] | 12.98 (2.81) |
| | 2017 | WW | Elixir | C | 237 (237) | 555 (1423) | 5: F(3), G(3), H(4) | 9.1 | 9 (0.82) |
| | 2018 | WB | California | – | 206.7 (318.7) | 415 (1917) | 3: F(3), G(1), H(2) | 7.8 | *1.78 (0.49)*[g] |

[a] SM: silage maize. GM: grain maize. CC: cover crop. WW: winter wheat. WR: winter rapeseed. SP: spelt. WB: winter barley. SB: spring barley. "Group" refers to variety specific grouping of cultivars with respect to similar properties. In winter wheat this mainly refers to protein content for elite winter wheat (E), quality wheat (A), bread wheat (B) and fodder wheat (C). For maize it is the FAO number.
[b] applied N fertilizer amounts were reported as commercial products by the farmers. Subsequently, $N_{tot}$, $NH_4-N$, $NO_3-N$ and $N_{amid}$ were calculated based on knowledge/estimates of respective N content in solid and liquid fertilizers. Numbers before brackets: fertilized amount between sowing and harvest and numbers in brackets fertilized amount between harvest of previous crop and harvest. The same was done for organic matter (OM).
[c] The total applied slurry (pig slurry – PS, cow slurry – CS – and biogas slurry – BS) was reported by the farmer. The organic matter content was subsequently calculated based on laboratory analyses in 2015 and 2016 for KR and 2014, 2015 and 2016 for research site EC6. Otherwise, average values were assumed based on expert knowledge. More details can be found in the management metadata file.
[d] H: herbicide. F: fungicide. I: insecticide. GR: growth regulator. The number before the colon indicates the number of times plant control measures were undertaken on the field between the harvest date of the previous crop and the harvest date of the current crop. The numbers in brackets show the total number of product groups applied, where several may have been used at one application time.
[e] Yield is reported as fresh mass of the generative biomass for all crops, except for SM, where it relates to fresh mass of the total aboveground biomass reported by the farmer. Numbers in brackets are the standard deviation of the replicate measurements. NA: not available.
[f] The low WR yield in 2013 was probably a consequence of damage due to hail.
[g] Values are much lower than expected.
[h] Last measured value before harvest was on 25 June (cf. biomass.csv).
[i] The silage maize yield at EC4 to EC6 was reported as dry mass.

**Table 4.** The instrumentation of the eddy-covariance stations (Wizemann et al., 2015) used until 2018. Occasional changes to the general layout are detailed in the text.

| Sensor | Manufacturer | Model | Measurement error (as given by the manufacturer) |
|--------|--------------|-------|--------------------------------------------------|
| **Aboveground sensors** | | | |
| 3D sonic anemometer, open-path infrared $H_2O/CO_2$ gas analyser (IRGA) | Campbell Scientific Inc. (UK) LI-COR Biosciences Inc. (USA) | CSAT3 LI-7500 | Horizontal: 1 mm s$^{-1}$ Vertical: 0.5 mm s$^{-1}$ $H_2O$ (rms): $\pm 3.3$ mg m$^{-3}$ at 10 Hz Pressure: $\pm 17$ hPa |
| Radiation, four-component* | Hukseflux Thermal Sensors (Netherlands) | NR01 | SW: $< 15$ W m$^{-2}$ at 1000 W m$^{-2}$ LW: $< 8$ W m$^{-2}$ at $-100$ W m$^{-2}$ LW$_{net}$ |
| Temperature, relative humidity | Vaisala Inc. (Finland) | HMP45 | Temperature: $\pm 0.3\,°C$, humidity: $\pm 2\%$ RH |
| Rainfall | Environmental Measurements Limited (UK) | ARG100 | $\pm 2\%$ |
| **Soil sensors** | | | |
| Temperature (up to five) | Campbell Scientific Inc. (UK) | 107 | $< \pm 0.3\,°C$, typically $\pm 0.1\,°C$ |
| TDR probes (up to five) | Campbell Scientific Inc. (UK) | CS616 | $< 1.5\%$ volumetric water content |
| Matric potential (up to five) | Campbell Scientific Inc. (UK) | Model 253 | |
| Heat flux plates (three) | Hukseflux Thermal Sensors (Netherlands) | HFP01 | $\pm 20\%$, typically $\pm 10\%$ |
| **Logger** | | | |
| Data logger | Campbell Scientific Inc., UK | CR3000, CR1000 | |

* SW: short wave, LW: long wave. TDR: time domain reflectometry.

**Table 5.** Installation depths (in cm) of the soil sensors. During selected periods, additional sensors were installed in greater depths, particularly at EC1 to EC3.

| Sensor | EC1 to EC3 | EC4, EC5 | EC6 |
|---|---|---|---|
| Temperature | 2, 6, 15, 30, 45 | 2, 6, 15, 30, 45 | 2, 6, 15, 30 |
| TDR | 5, 15, 30, 45, 75 | 5, 15, 30, 45 | 5, 15, 30 |
| Matric potential | 5, 15, 30, 45, 75 | 5, 15, 30, 45 | 5, 15, 30 |
| Soil heat flux | 8, three plates | 8, three plates | 8, three plates |

**Table 6.** Determined variables and description of the field cultivation data files (cultivation.csv) including farmer reported yield.

| Column name | Unit | Description |
|---|---|---|
| site | – | Field name |
| sdate | YYYY-MM-DDThh:mm | Sowing date |
| hdate | YYYY-MM-DDThh:mm | Harvest date |
| crop | – | Cultivated crop |
| var | – | Crop variety |
| code | – | Crop code (used throughout the database) |
| sdens | – | Seed density of sown crop |
| unit | $\text{seed m}^{-2}$; $\text{kg ha}^{-1}$ | Unit of seed density, sdens |
| yield | $\text{t ha}^{-1}$ | Yield as reported by the farmer |
| ref | DM o. FM CE18 | Reference mass of the yield (referenced to dry-matter weight of fresh matter weight) |
| residue | % | Percent of residues left after harvest |

**Table 7.** Determined variables and description of the soil management data files (soil_management.csv).

| Column name | Unit | Description |
|---|---|---|
| site | – | Field name |
| date | YYYY-MM-DDThh:mm | Date of soil management |
| Depth CE19 | m | Depth of soil management |
| Type | – | Type of soil management |
| Code | – | Abbreviation for Expert-N (Priesack, 2006) |

**Table 8.** Determined variables and description of the soil carbon and nitrogen measurement files (soil_cn.csv).

| Column name | Unit | Description |
|---|---|---|
| site | – | Field name |
| date | YYYY-MM-DDThh:mm | Measurement date |
| plot | – | Plot number |
| depth | cm | Indicator of soil layer depth: 30 for soil layer of 0–30 cm depth, 60 for soil layer of 30–60 cm depth and 90 for soil layer of 60–90 cm depth |
| bd | $\text{g cm}^{-3}$ | Bulk density of the soil sample |
| no3N | $\text{mg kg}^{-1}$ | Nitrate-N (no3N TS21) |
| nh4N | $\text{mg kg}^{-1}$ | Ammonium-N (nh4N) |
| soc | $\text{mg kg}^{-1}$ | Total soil organic carbon |
| son | $\text{mg kg}^{-1}$ | Total soil organic nitrogen |
| cmic | $\text{mg kg}^{-1}$ | Soil microbial carbon |
| nmic | $\text{mg kg}^{-1}$ | Soil microbial nitrogen |
| sub_plot_type | – | Vegetated: "veg"; bare plots in the years 2009, 2010 and 2012: "b09", "b10" and "b12" |

**Table 9.** Determined variables and description of the fertilizer data files (fertilization.csv).

| Column name | Unit | Description |
|---|---|---|
| site | – | Field name |
| date | YYYY-MM-DDThh:mm | Application date |
| FE_farm | – | Fertilizer as reported by the farmer |
| FE_com | – | Common name of fertilizer (full name) |
| FE_type | min; org | Mineral (min) or organic (org) fertilizer |
| FE_code | ssiii | Fertilizer code (as used by Expert-N; Priesack, 2006) |
| quantity | – | Quantity of applied fertilizer |
| unit | $\mathrm{kg\,ha^{-1}}$; $\mathrm{m^3\,ha^{-1}}$; $\mathrm{L\,ha^{-1}}$; $\mathrm{t\,ha^{-1}}$ | Unit of the applied fertilizer |
| DM | $\mathrm{kg\,ha^{-1}}$ | Calculated dry matter in the applied organic fertilizer |
| OM | $\mathrm{kg\,ha^{-1}}$ | Calculated organic matter in the applied organic fertilizer |
| N | $\mathrm{kg\,ha^{-1}}$ | Total quantity of N in the applied fertilizer |
| nh4N | $\mathrm{kg\,ha^{-1}}$ | Quantity of ammonium-N in applied fertilizer |
| no3N | $\mathrm{kg\,ha^{-1}}$ | Quantity of nitrate-N in applied fertilizer |
| amidN | $\mathrm{kg\,ha^{-1}}$ | Quantity of amidN CE20 in applied fertilizer |

The farmer-reported type and total amount of applied fertilizer type. Based on information provided from the fertilizer suppliers, analyses on the organic matter content of the organic fertilizers (slurry) and selected gap filling by expert knowledge, the dataset can be considered complete. However, it has to be acknowledged that the data on the organic fertilizers contain a non-quantified uncertainty. Further details on assumptions and calculations are given in Appendix A.

**Table 10.** Determined variables and description of the farmer-reported plant protection measures. The active substances and respective units were added based on expert knowledge (plant_protection.csv).

| Column name | Unit | Description |
|---|---|---|
| site | – | Field name |
| date | YYYY-MM-DDThh:mm | Application date |
| product | – | Name of product |
| type | – | Type of product: herbicide, fungicide, insecticide or growth control |
| dosage | – | Amount applied in units specified in units |
| unit_dos | $\mathrm{g\,ha^{-1}}$; $\mathrm{L\,ha^{-1}}$ | Unit of dosage |
| act_subst_1 | – | Names of active substance in the product |
| unit_subst_1 | $\mathrm{g\,L^{-1}}$; $\mathrm{g\,kg^{-1}}$ | Unit of act_subst1 in $\mathrm{g\,kg^{-1}}$ or $\mathrm{g\,L^{-1}}$ |
| act_subst_2 | – | Names of active substance in the product |
| unit_subst_2 | $\mathrm{g\,L^{-1}}$; $\mathrm{g\,kg^{-1}}$ | Unit of act_subst2 in $\mathrm{g\,kg^{-1}}$ or $\mathrm{g\,L^{-1}}$ |
| act_subst_3 | – | Names of active substance in the product |
| unit_subst_3 | $\mathrm{g\,L^{-1}}$; $\mathrm{g\,kg^{-1}}$ | Unit of act_subst2 in $\mathrm{g\,kg^{-1}}$ or $\mathrm{g\,L^{-1}}$ |
| comment | – | Due to inconsistencies in units reported by the farmer, comments to the interpretations of the reports were added |

**Table 11.** Determined variables and description of the weather data files (weather.csv).

| Column name | Unit | Description |
|---|---|---|
| site | – | Field name |
| date | YYYY-MM-DDThh:mm | Date of measurement |
| ws | $\mathrm{m\,s^{-1}}$ | Wind speed measured by a CSAT3 3D anemometer |
| ws_flag | – | Flag wind speed |
| wd | ° | Wind direction measured by a CSAT3 3D anemometer (degrees against north) |
| wd_flag | – | Flag wind direction |
| at | °C | Air temperature measured 2 m above ground |
| at_flag | – | Flag air temperature |
| rh | % | Relative humidity measured 2 m above ground |
| rh_flag | – | Flag relative humidity |
| ap | hPa | Atmospheric pressure |
| ap_flag | – | Flag atmospheric pressure |
| rs_down | $\mathrm{W\,m^{-2}}$ | Downwelling short-wave radiation (global radiation) |
| rs_down | – | Flag downwelling short-wave radiation |
| rl_down | $\mathrm{W\,m^{-2}}$ | Downwelling long-wave radiation |
| rl_down | – | Flag downwelling long-wave radiation |
| rs_up | $\mathrm{W\,m^{-2}}$ | Upwelling short-wave radiation (reflective radiation) |
| rs_up_f | – | Flag upwelling short-wave radiation |
| rl_up | $\mathrm{W\,m^{-2}}$ | Upwelling long-wave radiation |
| rl_up_f | – | Flag upwelling long-wave radiation |
| pr | mm | Precipitation measured 1 m above ground |
| pr_flag | – | Flag precipitation |

**Table 12.** Determined variables and description of the eddy-covariance measurement data files (flux_data.csv). For the variables nee_filtered, le_filtered and h_filtered data points were removed according to the following rule: for the respective quality flag > 6 and those measured values which were > 5 times the median of the previous 4 d were discarded. The Max Planck Institute (MPI) REddyProc R tool was used to gap-fill and for partitioning net ecosystem exchange (nee) into ecosystem respiration and gross primary productivity.

| Column name | Unit | Description |
|---|---|---|
| site | – | Field name |
| date | YYYY-MM-DDThh:mm | Date and time at the end of the 30 min averaging interval |
| nee | $mmol\,m^{-2}\,s^{-1}$ | Net ecosystem exchange of $CO_2$ |
| h | $W\,m^{-2}$ | Sensible heat flux |
| le | $W\,m^{-2}$ | Latent heat flux |
| rn | $W\,m^{-2}$ | Net radiation |
| shf1 | $W\,m^{-2}$ | Soil heat flux at 8 cm depth (heat flux plate 1) |
| shf2 | $W\,m^{-2}$ | Soil heat flux at 8 cm depth (heat flux plate 2) |
| shf3 | $W\,m^{-2}$ | Soil heat flux at 8 cm depth (heat flux plate 3) |
| ghf | $W\,m^{-2}$ | Ground heat flux: mean of shf1 to shf3 plus change in soil heat storage (dS) |
| dS | $W\,m^{-2}$ | Change in soil heat storage in the 0–8 cm layer and between $t_{i-1}$ and $t_i$ (calorimetric method) |
| qf_ustar | – | [a] Quality flag friction velocity (ustar), 1–9 |
| qf_h | – | [a] Quality flag sensible heat flux, 1–9 |
| qf_le | – | [a] Quality flag latent heat flux, 1–9 |
| qf_nee | – | [a] Quality flag net ecosystem exchange, 1–9 |
| r_err_ustar | % | Random-error friction velocity |
| r_err_h | % | Random-error sensible heat flux |
| r_err_le | % | Random-error latent heat flux |
| r_err_nee | % | Random-error net ecosystem exchange |
| noise_ustar | % | Instrumental-noise friction velocity |
| noise_h | % | Instrumental-noise sensible heat flux |
| noise_le | % | Instrumental-noise latent heat flux |
| noise_nee | % | Instrumental-noise net ecosystem exchange |
| z_l | – | Stability parameter |
| dir | ° | Wind direction (degrees against north) |
| ustar | $m\,s^{-1}$ | Friction velocity |
| nee_filtered | $mmol\,m^{-2}\,s^{-1}$ | Filtered net ecosystem exchange of $CO_2$ |
| le_filtered | $W\,m^{-2}$ | Filtered latent heat flux |
| h_filtered | $W\,m^{-2}$ | Filtered sensible heat flux |
| tair_gf | °C | [b] Gap-filled air temperature using REddyProc 1.2.2 |
| vpd_gf | hPa | [b] Gap-filled vapour pressure deficit using REddyProc 1.2.2 |
| rn_gf | $W\,m^{-2}$ | [b] Gap-filled net radiation using REddyProc 1.2.2 |
| nee_gf | $mmol\,m^{-2}\,s^{-1}$ | [b] Gap-filled net ecosystem exchange of $CO_2$ |
| h_gf | $W\,m^{-2}$ | [b] Gap-filled sensible heat flux |
| le_gf | $W\,m^{-2}$ | [b] Gap-filled latent heat flux |
| reco | $mmol\,m^{-2}\,s^{-1}$ | [c] Ecosystem respiration partitioned from measured nee |
| gpp_f | $mmol\,m^{-2}\,s^{-1}$ | [c] Gross primary production partitioned from measured nee and simulated reco |
| reco_dt | $mmol\,m^{-2}\,s^{-1}$ | [d] Ecosystem respiration partitioned from measured nee |
| gpp_dt | $mmol\,m^{-2}\,s^{-1}$ | [d] Gross primary production partitioned from measured nee |

[a] Gap-filling algorithm and [b] partitioning algorithm both from Reichstein et al. (2005). [c] Partitioning algorithm of Lasslop et al. (2010). [d] Foken (2006).

**Table 13.** Determined variables and description of the soil water content, temperature, heat storage and matric-potential measurements. Partially, the research sites had different numbers of sensors of a given type. All had temperature sensors installed at 2, 6, 15, 30 and 45 cm soil depth, and matric-potential and volumetric water content sensors were installed at 5, 15, 30, 45 and 75 cm depth, which are given separately in the individual files. For this reason, each data file (soil_site*_data.csv) has its own metadata file. * denotes the site number.

| Column name | Unit | Description |
| --- | --- | --- |
| site | – | Field name |
| date | YYYY-MM-DDThh:mm | Date of measurement |
| st | °C | Soil temperature measurements |
| mp | kPa | Matric-potential measurements |
| res | $k\Omega$ TS22 | Resistance of the matric-potential sensors |
| vwc | $m^3\,m^{-3}$ | Volumetric water content |
| wtt | µs | Wave travel time of the vwc sensor |
| vwc_cal | $m^3\,m^{-3}$ | Site-specific calibrated volumetric water content |
| Cv | $J\,cm^{-3}\,K^{-1}$ | Soil volumetric heat capacity, calculated from the vwc in 5 cm dry bulk density |
| dT | K | Change in near-surface soil temperature, calculated from the arithmetic mean of the 2 and 6 cm soil temperature recordings of two successive half-hourly time steps and computed differences of the mean |
| dS | $W\,m^{-2}$ | Change in soil heat storage in the 0–8 cm layer between two half-hourly time steps, calculated calorimetrically from Cv and dT (de Vries, 1963) |

* only reported for research sites EC1 to EC3 TS23

**Table 14.** Determined biomass variables and data description (biomass.csv).

| Column name | Unit | Description |
| --- | --- | --- |
| site | – | Field name |
| date | YYYY-MM-DDThh:mm | Measurement date |
| plot | – | Plot number of measurement |
| crop | – | Growing crop |
| plant_no | $\mathrm{m}^{-2}$ | Number of plants per square metre |
| tot_abv_bm | $\mathrm{g\,m}^{-2}$ | Total aboveground biomass: all plant tissues (stem + TS24 leaves) including generative biomass (grains) and glume or cob kernel and its leaves |
| veg_bm | $\mathrm{g\,m}^{-2}$ | Only vegetative, aboveground plant tissues: stem and leaves |
| gen_bm | $\mathrm{g\,m}^{-2}$ | Grains, cob kernels and seeds |
| ear | $\mathrm{g\,m}^{-2}$ | Ear weight |
| ear_no | $\mathrm{m}^{-2}$ | Ear number |
| glume | $\mathrm{g\,m}^{-2}$ | Glume weight |
| cob | $\mathrm{g\,m}^{-2}$ | Cob weight |
| cob_no | $\mathrm{m}^{-2}$ | Cob number per square metre |
| cob_leaves | $\mathrm{g\,m}^{-2}$ | Cob leaf weight |
| tsw | $\mathrm{g\,1000}^{-1}$ | Thousand seed weight |
| res_wc_tot | % | Residual water content (wt %) of total biomass samples dried at 105 °C |
| res_wc_veg | % | Residual water content (wt %) of vegetative-biomass samples dried at 105 °C |
| res_wc_glume | % | Residual water content (wt %) of glume biomass samples dried at 105 °C |
| res_wc_gen_bm | % | Residual water content (wt %) of generative-biomass samples dried at 105 °C |

**Table 15.** Determined variables and description of carbon and nitrogen content data of the crop biomass (cn.csv).

| Column name | Unit | Description |
| --- | --- | --- |
| site | – | Field name |
| date | YYYY-MM-DDThh:mm | Measurement date |
| plot | – | Plot number |
| crop | – | Growing crop |
| fraction | – | Fraction of the crop the C and N percentages are related to (straw = leaves and stem; generative = grains; total = (grains) + leaves + stem) |
| C | % | Nitrogen content in the respective fraction's biomass |
| N | % | Carbon content in the respective fraction's biomass |

**Table 16.** Determined leaf area index and data description (lai.csv).

| Column name | Unit | Description |
| --- | --- | --- |
| site | – | Field name |
| date | YYYY-MM-DDThh:mm | Measurement date |
| plot | – | Measurement plot number |
| crop | – | Growing crop |
| lai_mean | $m^2\,m^{-2}$ | Arithmetic mean of the measurement plot's leaf area index |
| lai_sd | $m^2\,m^{-2}$ | Standard deviation of the measurement plot's leaf area index |

**Table 17.** Determined plant development stage and height measurement and data description (phenology.csv).

| Column name | Unit | Description |
| --- | --- | --- |
| site | – | Field name |
| date | YYYY-MM-DDThh:mm | Measurement date |
| plot | – | Plot number |
| crop | – | Growing crop |
| replicate | – | Number of measurement within the plot |
| bbch | bbch | BBCH stage of the plant |
| plant_h | m | Height of the plant |

**Table 18.** Determined chamber flux measurements; the suffixes are identical to the ones in Table 9, as are the plot references here and in the data description.

| Column name | Unit | Description |
| --- | --- | --- |
| site | – | Field name |
| date | YYYY-MM-DDThh:mm | Measurement date |
| plot | – | Plot number |
| co2 | $kg\,C\,ha^{-1}\,h^{-1}$ | Soil $CO_2$ evolution |
| instrument | – | Instrument type |
| soil_temp | °C | Soil temperature at measurement |
| ambient_temp | °C | Ambient temperature at measurement |
| atmp | hPa | Atmospheric pressure at measurement |
| sub_plot_type | – | Vegetated: "veg"; bare plots in the years 2009, 2010 and 2012: "b09", "b10" and "b12" |

**Table 19.** Description of the attribute tables of the four GIS data model files which identify the location of the research areas, stations and plots in Tables 6–18 TS25. The main research plots are given in 03_research_plots.gpkg, identified as "veg" in the "sub_plot_type" column, and additionally "b09", "b10" and "b12" relate to the bare-soil plots in 04_research_plots_chambers.gpkg.

| Column name | Type | Description |
| --- | --- | --- |
| 01_research_sites.gpkg | | Location of the research sites |
| site | Integer | Research site/field number |
| field | String | Research site/field number |
| region | String | Identification of research region |
| 02_research_stations.gpkg | | Location of the eddy-covariance station |
| site | Integer | Research site/field number |
| station | String | |
| 03_research_plots.gpkg | | Location of the five plots per site for measurements of transient soil property and plant development |
| site | Integer | Research sites 1–6 used in data files to identify the research station |
| plot | Integer | Plot numbers of the soil property and plant development observations in Sect. 2.3.3 and 2.3.4 |
| 04_research_plots_chambers.gpkg | | Location of all plots, including the those in Sect. 2.3.5 |
| plot | Integer | Plot numbers of the soil property and plant development observations in Sect. 2.3.3–2.3.5 |
| sub_plot_type | String | Vegetated: "veg"; bare plots in the years 2009, 2010 and 2012: "b09", "b10" and "b12" |

## Appendix A

**Table A1.** Further information on the quantification rules to calculate the amount and type of mineral N from the reported applied mineral fertilizers on the 54 site years and the organic matter content.

| Fertilizer type | Density $\mathrm{kg\,L^{-1}}$ | $N_{total}$ % | $NO_3-N$ % | $NH_4-N$ % | $N_{amid}$ % | Type – |
|---|---|---|---|---|---|---|
| Ammonium sulfate solution | 1.25 | 15 | 3.5 | 8.6 | 2.9 | Liquid |
| Ammonium nitrate urea solution | 1.28 | 28 | 7 | 7 | 14 | Liquid |
| PIASAN | 1.31 | 25 | 5 | 9 | 11 | Liquid |
| Calcium ammonium nitrate | n/a | 27 | 13.5 | 13.5 | n/a | Solid |
| NPK CE21 | n/a | rv | $0.5 \times N_{total}$ | $0.5 \times N_{total}$ | n/a | Solid |
| Urea/ALZON | n/a | rv or 46 | n/a | n/a | rv or 46 | Solid |
| Ammonium sulfate nitrate | n/a | 21 | 15.5 | 5.5 | n/a | Solid |
| PIAMON 33 S | n/a | 33 | n/a | 10.4 | 22.6 | Solid |
| InnoFert | n/a | 24 | 7.8 | 16.2 | n/a | Solid |

n/a: TS26 not applicable. rv: reported value.

**Table A2.** Further information on the quantification rules to calculate the amount and type of mineral N from the reported applied mineral fertilizers on the 54 site years and the organic matter content.

| Fertilizer type | TS % | OM % | $N_{total}$ % | $NH_4-N$ % |
|---|---|---|---|---|
| Biogas slurry | 6 | 73 | 0.43 | 0.26 |
| Cow slurry | 8 | 71 | 0.39 | 0.21 |
| Pig slurry | 5 | 71 | 0.56 | 0.42 |

TS: total solids. OM: organic matter.
Nutrient contents determined in the laboratory in 2015 and 2016 for Kraichgau and 2014, 2015 and 2016 for research site EC6 (SA). Otherwise, average values were assumed based on expert knowledge, as given below.

Earth Syst. Sci. Data, 13, 1–30, 2021 https://doi.org/10.5194/essd-13-1-2021

Please note the remarks at the end of the manuscript.

## Appendix B

**Table B1.** Average yields of the Kraichgau district Enzkreis for the years 2010–2018. Values that are not available are indicated by NA. Data from the Statistisches Landesamt Baden-Württemberg accessible at https://www.statistik-bw.de/SRDB/ (last access: 20 May 2020): Regionaldaten – Land- und Forstwirtschaft – Ernte – Hektarerträge der Feldfrüchte (Landkreis Enzkreis).

| Kraichgau-Enzkreis | 2010 | 2011 | 2012 | 2013 | 2014 | 2015 | 2016 | 2017 | 2018 | Average |
|---|---|---|---|---|---|---|---|---|---|---|
| | | | | | $\mathrm{Mg\,ha^{-1}}$ | | | | | |
| Winter wheat | 6.72 | 7.74 | 7.38 | 7.35 | 7.08 | 8.3 | 6.62 | 7.78 | 8.04 | 7.4 |
| Winter barley | 6.45 | 6.1 | 6.92 | 6.82 | 7.04 | 7.05 | 6.21 | 7.46 | 7.17 | 6.8 |
| Spring barley | 5.03 | 5.36 | 6.28 | 4.84 | NA | 5.35 | NA | NA | NA | 4.5 |
| Oat | 4.53 | NA | NA | NA | NA | 5.13 | NA | NA | NA | 4.8 |
| Triticale | 5.66 | NA | NA | NA | NA | NA | NA | NA | NA | 5.7 |
| Grain maize | 9.54 | NA | NA | NA | NA | NA | 8.82 | 10.34 | 9.93 | 9.7 |
| Potatoes | 31.51 | NA | NA | NA | NA | NA | NA | NA | NA | 31.5 |
| Sugar beat | 0 | NA | NA | NA | NA | NA | NA | NA | NA | NA |
| Winter rapeseed | 3.8 | 3.3 | 3.5 | 3.7 | 5.2 | 4.5 | 3.6 | 4.3 | 4.2 | 4.0 |
| Silage maize | 43.9 | 56.9 | 51.4 | 42.1 | 57.6 | 42.7 | 42.3 | 43.3 | NA | 47.5 |

**Table B2.** Average yields of the Swabian Alb district Alb-Donau for the years 2010–2018. Values that are not available are indicated by NA. Data from Statistisches Landesamt Baden-Württemberg accessible at https://www.statistik-bw.de/SRDB/ (last access: 20 May 2020): Regionaldaten – Land- und Forstwirtschaft – Ernte – Hektarerträge der Feldfrüchte (Landkreis Enzkreis).

| Swabian Alb | 2010 | 2011 | 2012 | 2013 | 2014 | 2015 | 2016 | 2017 | 2018 | Average |
|---|---|---|---|---|---|---|---|---|---|---|
| Alb-Donau-Kreis | | | | | $\mathrm{Mg\,ha^{-1}}$ | | | | | |
| Winter wheat | 7.2 | 8.1 | 7.6 | 7.97 | 8.56 | 7.95 | 7.11 | 8.09 | 8.13 | 7.9 |
| Winter barley | 6.21 | 7.18 | 7.22 | 7.13 | 7.68 | 6.78 | 6.77 | 7.44 | 6.83 | 7.0 |
| Spring barley | 5.51 | 6.23 | 6.87 | 5.86 | 6.35 | 5.61 | 5.68 | 5.28 | 5.76 | 5.9 |
| Oat | 5.07 | 4.56 | 6.35 | 5.29 | 5.44 | NA | 4.84 | 5.04 | NA | 5.2 |
| Triticale | 6.95 | 8.13 | 7.75 | 8.28 | 8.75 | 7.67 | 6.19 | 7.55 | 7.03 | 7.6 |
| Grain maize | 10.04 | 10.85 | 11.31 | 10.38 | NA | 9.04 | 9.52 | 8.2 | 8.79 | 9.8 |
| Potatoes | 34.62 | 41.01 | 55.88 | 25.65 | NA | 38.56 | 46.49 | 42.05 | 37.9 | 40.3 |
| Sugar beat | 71.06 | 75.3 | 79.33 | NA | NA | NA | 65.19 | NA | 65.46 | 71.3 |
| Winter rapeseed | 3.76 | 3.46 | 4.19 | 4.04 | 5.1 | 4 | 3.87 | 3.94 | 4.14 | 4.1 |
| Silage maize | 45.81 | 46.58 | 50.27 | 41.79 | 48.56 | 38.13 | 48.01 | 47.54 | 45.16 | 45.8 |

**Table B3.** Average yields of the Swabian Alb district Reutlingen for the years 2010–2018. Values that are not available are indicated by NA. Data from Statistisches Landesamt Baden-Württemberg accessible at https://www.statistik-bw.de/SRDB/ (last access: 20 May 2020): Regionaldaten – Land- und Forstwirtschaft – Ernte – Hektarerträge der Feldfrüchte (Landkreis Enzkreis).

| Swabian Alb Reutlingen | 2010 | 2011 | 2012 | 2013 | 2014 | 2015 | 2016 | 2017 | 2018 | Average |
|---|---|---|---|---|---|---|---|---|---|---|
| | | | | | $Mg\,ha^{-1}$ | | | | | |
| Winter wheat | 6.29 | 6.53 | 6.29 | 5.51 | 7.19 | 5.92 | 5.87 | 6.64 | 7.06 | 6.4 |
| Winter barley | 6.06 | 6.38 | 5.82 | 5.5 | 7.69 | 5.97 | 6.33 | 6.93 | 6.74 | 6.4 |
| Spring barley | 4.87 | 5.31 | 5.38 | 4.39 | 6.35 | 4.56 | 3.76 | 4.95 | 5.26 | 5.0 |
| Oat | 4.72 | 5.3 | 5.82 | 5.1 | 5.9 | 4.62 | 4.92 | 3.99 | NA | 5.0 |
| Triticale | 6.4 | 6.71 | 7.3 | 6.73 | 6.38 | 6.14 | 5.19 | 6.15 | 6.67 | 6.4 |
| Grain maize | NA | NA | NA | NA | NA | NA | NA | NA | NA | NA |
| Potatoes | 28.2 | 31.85 | 27.49 | 20.33 | 29.39 | 15.4 | 7.03 | 23.25 | 27.21 | 23.4 |
| Sugar beat | NA | NA | NA | NA | NA | NA | NA | NA | NA | NA |
| Winter rapeseed | 3.65 | NA | 3.89 | 2.88 | NA | NA | NA | 3.48 | 3.84 | 3.5 |
| Silage maize | 35.81 | 41.46 | 41.55 | 35.85 | NA | 31.48 | 32.91 | 35 | 34.93 | 36.1 |

**Supplement.** The supplement related to this article is available online at: https://doi.org/10.5194/essd-13-1-2021-supplement.

**Author contributions.** The project was conceptualized (ideas, formulation or evolution of overarching research goals and aims) by GC, PH and TS. The data were curated (management activities to annotate, i.e. produce metadata, and scrub data and maintain research data, including software code, where necessary for interpreting the data itself, for initial use and later re-use) by TW, CT, HDW, PH, ML, YFN, MSD, AP, PK and JI. The formal analysis was completed (application of statistical, mathematical, computational or other formal techniques to analyse or synthesize study data) by TW, HDW, PH, AP and PK. The funding was acquired (acquisition of the financial support for the project leading to this publication) by TM, AF, GC, PH, JI, VW and TS. The investigation was completed (conducting a research and investigation process, specifically performing the experiments or data/evidence collection) by KB, HDW, TW, RE, PH, YFN, ML, MSD, AP, PK and TS. The methodology was developed (development or design of methodology and creation of models) by TM, PH, TS and JI. The project was administered (management and coordination responsibility for the research activity planning and execution) by PH, VW, TS and JI. The resources were acquired (provision of study materials, reagents, materials, patients, laboratory samples, animals, instrumentation, computing resources or other analysis tools) by TM, GC, PH, AF, JI and TS. The software was managed (programming and software development, computer programme design, computer code and supporting algorithm implementation, and existing code component testing) by TW, GC, HDW and JI. The project was supervised (oversight and leadership responsibility for the research activity planning and execution, including mentorship external to the core team) by TM, GC, PH, AP, JI, VW and TS. The results were validated (verification, whether as a part of the activity or separate, of the overall replication/reproducibility of results/experiments and other research outputs) by TW, CT, KB, PH, AP, PK and JI. The results were visualized (preparation, creation and/or presentation of the published work, specifically visualization/data presentation) by TW, PH and JI. The original draft was written (preparation, creation and/or presentation of the published work, specifically writing the initial draft, including substantive translation) by TW, PH, YFN, ML, AP and JI. The writing was reviewed and edited (preparation, creation and/or presentation of the published work by those from the original research group, specifically critical review, commentary or revision, including during the pre- or post-publication stages) by TW, TM, CT, AP, JI, PH, TS and MSD. CE22

**Competing interests.** The contact author has declared that neither they nor their co-authors have any competing interests.

**Disclaimer.** Publisher's note: Copernicus Publications remains neutral with regard to jurisdictional claims in published maps and institutional affiliations.

**Acknowledgements.** We specifically acknowledge the distinguished farmers who enabled this research, namely the elder Mr. TS27 Bosch (deceased) and the younger Mr. TS28 Bosch (EC1 to EC3), Mr. TS29 Fink (EC4), Mr. TS30 Hermann (EC5), and Mr. TS31 Reichert (EC6), without whom the collection of this dataset would have not been possible. This dataset is the result of the DFG (German Research Foundation) integrated project PAK 346 Structure and Functions of Agricultural Landscapes under Global Climate Change – Processes and Projections on a Regional Scale and the DFG-funded Research Unit 1695 Agricultural Landscapes under Global Climate Change – Processes and Feedbacks on a Regional Scale. TW was funded by the Collaborative Research Center 1253 CAMPOS (project 7: Stochastic Modeling Framework of Catchment-Scale Reactive Transport CE23), funded by the German Research Foundation (DFG, grant agreement SFB 1253/1 2017). Further, we thank the technical staff team members B. TS32 Prechter and T. TS33 Schreiber and student helpers F. TS34 Baur and C. TS35 Schade. KB was financed by a scholarship in the frame of the Erasmus Mundus IAMONET-RU programme. The authors are grateful to W. TS36 Damsohn, G. TS37 Gensheimer, Z. TS38 Kauf, Mr. TS39 Strohm and the students for assistance with the field experiments. Parts of this study were funded by the Federal Ministry of Education and Research (grant no. 01PL11003) for the Humboldt Reloaded projects at the University of Hohenheim, Germany. Lastly, we thank A. TS40 Klumpp and J. TS41 Auerbacher for their support.

**Financial support.** This TS42 research has been supported by the Deutsche Forschungsgemeinschaft (grant no. 193709899).

**Review statement.** This paper was edited by Birgit Heim and reviewed by two anonymous referees.

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

## Remarks from the language copy-editor

## Remarks from the typesetter

TS19   Please note that the range 7–19 has been changed to 6–18 due to changes in the Author contributions section.

TS20   This figure is not mentioned within the text. Please confirm its position.

TS21   Please check no3N and nh4N. Should be these expressions formatted chemically.

TS22   Please note that the unit has been changed to the physical standard unit.

TS23   Please check. This footnote can not be found in the table above.

TS24   Please check. If you mean "and", please write it out in full. "+" is only allowed in mathematical terms.

TS25   Please note that the range 7–19 has been changed to 6–18 due to changes in the Author contributions section.

TS26   Please note our standard abbreviation.

TS27   Please provide the full name. Also note that academic degrees are not supposed to be mentioned here.

TS28   Please provide the full name.

TS29   Please provide the full name.

TS30   Please provide the full name.

TS31   Please provide the full name.

TS32   Please provide the full name.

TS33   Please provide the full name.

TS34   Please provide the full name.

TS35   Please provide the full name.

TS36   Please provide the full name.

TS37   Please provide the full name.

TS38   Please provide the full name.

TS39   Please provide the full name.

TS40   Please provide the full name.

TS41   Please provide the full name.

TS42   Please note that there is a discrepancy between funding information provided by you in the acknowledgements and the funding information you indicated during manuscript registration, which we used to create this section. Please double-check your acknowledgements to see whether repeated information can be removed from the acknowledgements or changed accordingly. If further funders should be added to this section, please provide the funder names and the grant numbers. Thanks.

TS43   Please provide the publisher and place of publication (city).

TS44   Please provide the publisher and place of publication (city).

TS45   Please provide place of publication (city).

TS46   Please provide the publishing journal or institution.

TS47   Please provide place of publication (city).

TS48   Please update.

TS49   Please provide the publisher and place of publication (city).

TS50   According to paper at this DOI, this should be "Mualem". Please check.

TS51   Please provide the publisher and place of publication (city).

TS52   Please provide the following information: day/month/year.

TS53   Please provide the publisher/repository.