# Peer review of "Multi-site, multi-crop measurements in the soil-vegetation-atmosphere continuum: A comprehensive dataset from two climatically contrasting regions in South West Germany for the period 2009-2018"

_Earth System Science Data, 2020_

## Author Comment (AC1)

Dear authors,

field observations are the inevitable basis of model investigations as the foundations of calibration and validation. They also form the ground truth for the development of remote sensing applications. Both has the potential for better results with more high quality field data being available. Therefore, for this kind of datasets, originality is not an important criterion. The manuscript under review contributes a large and comprehensive dataset covering relevant observations. The dataset is unique, useful and complete, which make it suitable for publication in ESSD. Most of the methods of data collection described follow community standards. The dataset is well organized in appropriate file formats. The data files are well documented in metadata files and tables. The text is well written with only very few language errors and the list of references is complete.

> Dear reviewer, we thank you for your time and effort in reviewing the manuscript, metadata files and data set. We also thank you for your strong support of our manuscript and, particularly, as you say, in evaluating it as a useful and complete dataset.
>
> Below, we have taken great care to address your comments and questions in a detailed point-by-point manner. We have followed almost all of your suggestions, and as a consequence have made changes to the manuscript. We are convinced this has improved the manuscript in structure and completeness. Changes are marked by tracked changes, and in our detailed responses below, changes in the text are highlighted by italic font. The location of changes to the text are in relation to the text with track changes.
>
> Lastly, we would like to add that we value this as excellent as dedicated and intensive review which remains enthusiastic and supportive throughout. This is very much appreciated, thanks!

However, I see some problems with the structure of the text. Resolving these will improve the usability for readers. The current structure of the text makes it more difficult to follow than necessary. I do mainly refer to the differentiation of "field measurements", "field sampling", and "laboratory measurements", which causes data on the same compartments of the system (soil, plants) to be described in more than one place. For a user of the data, it makes it easier to find required information, if all soil data and all vegetation data is described in the same section or in consecutive sections, respectively. Please find additional comments on this in the specific comments.

> AGREE. We have restructured parts of the text, mostly by copy+pasting and thereby follow the suggestions from both reviewers. These changes become apparent in the responses to the individual comments.

Furthermore, the text does not cover all of the data provided in the dataset (for1695_data.zip, the file containing the photos, and the supplementary data):

1. Soil respiration measurements are in the dataset as is metadata for it. It is also mentioned on the text as chamber measurements or soil respiration. However, the description of the measurement process, devices applied, and so on, is missing.
2. gis_data_model: this is a valuable supplement to the data. It deserves to be mentioned in the text.
3. Soil profile data: This should be explicitly addressed in the text.
4. The soil water figures in the supplementary material may be mentioned in the appropriate position.

ad 1) AGREE. We have now greatly extended this in section 2.3.2 and are convinced this is sufficient to understand the measurement process (LL xxx):

*"On selected sites in some additional measurement campaigns to determine, soil surface CO2 flux with chamber measurements were was done on taken in both bare fallow and vegetated plots. with EGM-2 and EGM-4 CO2 detectors (PP Systems, Amesbury, MA) (Demyan et al., 2016). Both types of soil respiration plots (both the bare and vegetated) were located close to three plots used for plant observations/biomass harvests. Since bare plots were maintained through multiple years of the experiment, the plots were located at the periphery of the fields to be outside of the EC footprint. Each of the three plots was allocated randomly within a third of the field outside the main EC footprint avoiding field edges, tractor pathways, and other non-representative spots. Soil surface CO2 flux was measured via portable infrared gas analyzers (EGM-2 and EGM-4, PP Systems Amesbury, Massachusetts, USA) with attached soil temperature probe and soil respiration chamber (Demyan et al., 2016; Laub et al., 2021). The soil respiration flux chamber was 10 cm in diameter with an internal volume of 1171 $cm^{-3}$. Fluxes were measured during the growing period of different years (2009, 5×; 2010, > 15×; 2014, 3×; 2015, 7×) and an intensive approximately weekly campaign May 2012 to June 2013, 40×). Six replicate measurements were taken within each subplot (vegetated and bare fallow) during each measurement day. Measurement order of the plots was randomized each day to avoid time of day effects. Individual measurements which were greater than six times the yearly median value were removed as outliers."*

We included additional references.

Demyan, M.S., Ingwersen, J., Funkuin, Y.N., Ali, R.S., Mirzaeitalarposhti, R., Rasche, F., Poll, C., Müller, T., Streck, T., Kandeler, E., Cadisch, G., 2016. Partitioning of ecosystem respiration in winter wheat and silage maize—modeling seasonal temperature effects. Agriculture, Ecosystems & Environment 224, 131–144. doi:https://doi.org/10.1016/j.agee.2016.03.039

Laub, M., Ali, R.S., Demyan, M.S., Nkwain, Y.F., Poll, C., Högy, P., Poyda, A., Ingwersen, J., Blagodatsky, S., Kandeler, E., Cadisch, G., 2021. Modeling temperature sensitivity of soil organic matter decomposition: Splitting the pools. Soil Biology and Biochemistry 153, 108108. doi:https://doi.org/10.1016/j.soilbio.2020.108108

ad 2) AGREE. It is now mentioned in section 3 (LL xxx):

A new table to describe the columns of the attribute tables of the **gis data model** was included:

**Table 1: Description of the attribute tables of the four GIS data model files which identify the location of the research areas, stations, and plots in Tables 7-19. The main research plots are given in *03_research_plots.gpkg*, identified as "veg" in the *sub_plot_type* column, and additional "b09", "b10", "b12" relate to the bare soil plots in *04_research_plots_chambers.gpkg*.**

| Column Name | Type | Description |
|---|---|---|
| *01_research_sites.gpkg* | | *location of the research sites* |
| site | integer | research site/field number. |
| field | string | research site/field number |
| region | string | identification of research region |
| *02_research_stations.gpkg* | | *location of eddy-covariance station* |
| site | integer | research site/field number. |
| station | string | |
| *03_research_plots.gpkg* | | *location of the five plots per site for transient soil property and plant development measurements* |
| site | integer | research sites 1-6 used in data files to identify the research station. |
| plot | integer | plot numbers of the soil property and plant development observations in section 2.3.3 and 2.3.4 |
| *04_research_plots_chambers.gpkg* | | *location of all plots, including the those in section 2.3.5.* |
| plot | integer | plot numbers of the soil property and plant development observations in section 2.3.3-2.3.5 |

ad 3) AGREE. We added a sentence (P LL xxx):
*"Physico-chemical properties of the soils are provided in **Error! Reference source not found.**."*

ad 4) AGREE. We have now referenced them in the text (LL xxx).

Vegetation green fraction, on the contrary, is described in the text but not included in the dataset.  This also requires correction. Please see my specific comments on this.

AGREE. Addressed in the specific comment below.

There is only one major deficit of the dataset and the manuscript: There is not sufficient information on data quality. The text lacks descriptions of measures for quality control and assurance as applied during the measurement campaigns and subsequent analyses in the lab or during data analysis. Furthermore, to make the data usable, information on errors and uncertainty need to be presented. This comprises measurement errors of devices used (there is some data included on this aspect), or errors resulting from handling samples in the field and in the lab. For each value in the dataset there has to be a number or a statement on uncertainty. Such a statement may also be an explanation, why no quantitative estimate of uncertainty is given. It has also to be ensured, that flags included in the data files are explained in the text or in the metadata files.

AGREE. We introduced a new subsection 2.5. dedicated to the quantitative and qualitative description of errors and uncertainties

Another information, which is of interest to readers and potential users of the data, is the fraction of missing values. Please consider adding this.

DISAGREE. We think there is no easily accessible and suitable preparation for this information which truly provides value. Moreover, the data format and dataset size in kB is so small, this can be quickly done by the potential users. Also, we provided some figures, which give a good graphical representation of the data. Lastly, many of the measured data time series do not permit to evaluate the fraction of missing values, as the measurement process was in fact not scheduled regularly (e.g. lai.csv, biomass.csv, …). Most importantly, in the EC and weather data, gap filling was used profusely.

Overall, the manuscript is a good text describing an impressive extensive data set. I am looking forward to the revised version of the manuscript, which will be a valuable contribution even beyond the modellers community.

Thanks!

**Specific comments**

1.  P2L7: The EC data was aggregated to 30 min.

AGREE. We have included a respective sentence (LL xxx):

"The EC data were logged in 10 Hz resolution on a CR3000 data logger (Campbell Scientific Inc., Logan, UT, USA). *The EC data was aggregated to 30 min (raw data available upon request).* All other sensor data were stored in 30 min intervals."

2. Fig1: The conceptual model seems not completely coherent. Why are crops shown in in the "land surface" compartment when there is a "crops" compartment? Does the land surface include non-crop vegetation in this conceptual model? And does the crop compartment have its own soil model? Doesn't the crop model simulate plant water uptake, which in turn has to be passed to the soil hydrology model (brown arrow from right to left)? Please make this coherent or comment on it.

[Figure]

AGREED. We have modified it, see below. (left) the old version, (right) the new version.

The red ellipsoids are inserted to ease the review and are not part of the figure.

3. P4L15ff: Do not forget the weather data.

AGREE. Well spotted. We included a sentence with the required details (LL xxx):

"[…] aggregated to 30 min resolution (from 2009 to 2016 fluxes were not measured during the winter months), *iv) additionally, meteorological data collected at 30 min resolution comprising rain, air temperature at two meters height, and relative humidity*, v) soil characteristics, vi) soil state measurements including water content, temperature, and matric potential (30 min), the soil […]"

4. Fig2: Please include a (small) map showing where Baden-Württemberg is located in Europe for the international readers.

DISAGREE. We think it is clear that Baden-Württemberg is part of Germany, and the map clearly shows were France, Switzerland, and Austria are located using the international accepted naming convention.

5. Sections 2.1 and 1: Consider moving the information on the sites from the introduction to the site description and the information on funding to the introduction

AGREE. An excellent suggestion.

1) We moved the following text to the introduction (now in LL xxx):
"Field research was part of the two wider integrated research projects PAK 346 *Structure and Function of Agricultural Landscapes under Global Climate Change – Processes and Projections on a Regional Scale* and RU 1695 *Agricultural Landscapes under Global Climate Change – Processes and Feedbacks on a Regional Scale*, funded by the German Research Foundation (DFG)."
2) Following the suggestion, the **Material and Methods** section now starts before „Both research areas were intensively used agricultural landscapes; 1) […]".

6. Section 2.1.1 and 2.1.2: The source of the climatic data is missing for KR if Troost and Berger is the source for SA.

AGREE. We deleted the reference. It was out of place.

7. Have the sites been used as agricultural fields for a long time? This is interesting for readers from areas where agriculture has begun only some hundred years ago and the soil's slow carbon pools are still influenced by the conversion.

AGREE. Excellent suggestion. We have included this information in subsubsection 2.1.1 (Kraichgau, EC1-3), and in subsubsection 2.1.2 (Swabian Alb, 2.1.2)

*"Photos show that the EC1-3 were meadows until the 1960s after which the area at large was drained, and agricultural fields were established. "*

*„Both EC4 and EC5 have been used as agricultural fields since the 1970s. It is likely that they were subject to land consolidation in the 1980s."*

*"Based on personal accounts, EC6 is known to have been under agricultural operation since at least the 1940s. Before the land consolidation in 1987, the field was separated into at least 20 different plots. The current owner has been using the field at EC6 since 1993. "*

8. How are soils in SA classified following WRB?

AGREED. This was missing. We have now included this information in subsubsection 2.1.1

*"The soils are classified as a Calcic Luvisol at research site 4, Anthrosol at site 5, and Rendzic Leptosol at site 6 according to the (World Reference Base for Soil Resources, WRB; Michéli et al., 2006)."*

9. Table3: is this table required? Which additional information is given compared to the data tables? If none, consider removing it. Footnotes 6 to 8 are relevant to users of the data. Consider adding this in a comment column in the data file.

DISAGREE.

1) While the data tables (.csv files) are long format, Table 3 is a comprehensive and compact overview. Since these are agricultural ecosystems, we are convinced that the characterisation of the land use is highly relevant for users when considering if this is the right kind of ecosystem to study.

2) We have no comment sections in the data tables (.csv files), and prefer not to include this, for these single and individual data points.

10. The information on NA should be in the caption.

AGREE. And it already was ☺

11. Please put the line explaining the asterisk in a separate row in the footnotes. I had to search for it.

AGREE. Done.

12. What are the numbers in parentheses in the yield per plot column?

AGREE. It was not clear. Footnote 5 in Table 3 now includes the required explanation *"Numbers in brackets are the standard deviation of the plot replicates in the field."*

13. Section2: I am missing a section on soil property data. This has been described above but since these are also field measurements (in the same sense as crop measurements are), they should be described here.

AGREE. We now refer to section 2.3.3. in the Table 1 caption, where the details are now given.

14. Section 2.2: The information on composition of fertilizers in the metadata file is valuable. It should be advertised in the text.

AGREE. Thanks for this excellent idea. We included a reference to Table A1-2 in section 2.2.:

*"Physico-chemical properties of the fertilisers are given in Table A1-2 and were used to calculate the input of nitrogen and organic matter to the fields."*

15. Why is crop yield assigned to management and not to crop/vegetation data?

NO ACTION taken. There are two yields reported. Those by the farmer, which are included in the management file where yield is given in the field index cards by the farmer, and those determined by observation.

For clarity: We now have added a paragraph to explain this *verbatim* in section 2.2.

*"Note that the yield values reported in last two columns of Table **Error! Reference source not found.** differ from each other. In the field column values are farmer reported yields for the entire field. These are affected by harvest losses, no yields on tractor tracks, and reduced yields due to side effects on the field. In the plot column the reported values stem from observations on experimental plots located far away from the edge of the field and between tractor tracks. This explains why the farmer values are mostly smaller than the plot values."*

16. L12f: Is there an explanation for the discrepancies? What is the uncertainty of the different numbers?

AGREE partly. We have now included a paragraph with details in section 2.2.

*"Note that the yield values reported in last two columns of Table **Error! Reference source not found.** differ from each other. In the field column values are farmer reported yields for the entire field. These are affected by harvest losses, no yields on tractor tracks, and reduced yields due to side effects on the field. In the plot column the reported values stem from observations on experimental plots located far away from the edge of the field and between tractor tracks. This explains why the farmer values are mostly smaller than the plot values. Values in brackets are standard deviations over the plots. For silage maize in the Kraichgau region, the farmer values are reported as fresh mass."*

17. Please state, how the positions of the five plots per site (field) were chosen

AGREE. At the end of the introduction to section 2, we now state that

*"The location of the research stations and plots followed practical considerations."*

18. Also refer to the tables in appendix A.

AGREE. Well spotted. We now do this by writing (LL xxx):
*„Physico-chemical properties of the fertilisers are given in Table A1-2, and were used to calculate the input of nitrogen and organic matter to the fields."*

19. Section 2.3.1: The data files include columns showing flags. The flagging concept needs to be explained here.

AGREE. Done. Also in the corresponding metadata.

*„The weather data gap filling and flags were done using an automated Fortran program, which we summarise, here. For all variables no gap filling is marked by flag 0. Gap filling was first tried by using data from an adjacent station. The gap filled data was then flagged as 11, 12, 13, 14, 15, 16, for data from EC1 through EC6, respectively. If no wind speed or wind direction data from the adjacent stations was available, a random wind speed was sampled from the data of the previous twelve hours (flag = 6). Air temperature was filled by linear interpolation for if the data gap was no more than three measurements, and correspondingly flagged by 1-3. In other cases, data from neighbouring stations was used. The humidity data was treated in the same way as air temperature, but if a missing value is >99% the gap is filled with 99.9 (flag=5). The air pressure data gaps were also filled like the air temperature. If not data from neighbouring stations are available, either, the pressure is set to the average pressure of the region (flag=7). For the downwelling (down) and upwelling (up) shortwave radiation (rs) and longwave radiation (rl) the following gap filling approach was done: For rs_down and rs_up, data gaps are filled with data from the neighbouring stations, For rs_up, data points are computed based on the albedo of the previous dataset and rs_down (flag=4). rl_up was filled in the same way as air temperature. Additionally, rl_up values were checked for plausibility. If values were below 200 W m$^{-2}$, gaps were filled up with data. Precipiation data was gap filled in the same way as rs_down."*

20. Section 2.3.2: The caption of Figure 4 mentions gap-filled data. If gap-filling was done, it needs to be described. Otherwise, the data not included in the dataset should not be shown in the figure.

AGREE. Thus, we make it clear by being precise in (LL xxx):
*„For this and for the gap-filling, we used the R package REddyProc (Wutzler et al., 2020)."*

21. Section 2.3.3: incomplete sentence

AGREE. We now write (changes in italics):

*„The remaining sites and years are presented in the supplementary information. "*

22. The metadata files mention a calibration step for the soil moisture. This should be mentioned in the text.

AGREED. This was rather curt. We have no specified in section 2.3.3 that

*"At EC1-3, the soil water content sensors were calibrated to insitue gravimetric soil water content data. In EC4-6, only the factory calibrated time series are provided."*

23. A reference to Table 5 is missing.

DISAGREE. This is not true, the text does reference the table.

24. Section 2.3.4: The heading of the section is inappropriate since it also reports on biomasses and height.

AGREE. Sections 2.3.4 is now renamed to "*Plant sampling and development variables*"

25. The dataset does not include fresh biomass or vegetation water content. This is valuable information for e.g. radar remote sensing applications. If this data has been measured in an appropriate way, it needs to be added to the dataset. The section also lacks information on how biomass was dried, how much of the harvested biomass was used for LAI determination. Sometimes, blossoms are assigned to generative biomass and only harvestable biomass is reported instead of generative biomass. How was this done in this case?

We think this comment deals with four points

    1) water content in biomass and vegetation
    2) LAI determination
    3) Discrimination between generative and vegetative biomass
    4) Method to determine dry weight of the biomass.

ad 1) AGREE. For most measurements this information is not available as no measurements were made. An exception are selected and reported measurements during 2017 and 2018. Since a good point was risen, and to maximise the potential of future datasets, we included a statement in the conclusions as a call for including this data in future datasets (LL xxx):

> *"We also recommend to regularly include measurements on the water content of the crops for remote sensing applications."*

ad 2) DISAGREE. In section 2.3.4, we write that "A LAI-2000 Plant Canopy Analyzer (LI-COR Biosciences Inc., USA) was used to measure total leaf area index" was used. I think this unmistakingly shows, that we did not analyse LAI on harvested leaves.

ad 3) We now give this detail in the text in section 2.3.3.:

> *"Generative biomass for winter wheat, spelt and barley is only the grain, for the maize it is the spindle, and for winter rapeseed only the seeds without the pods. The remaining parts of the plants are considered as above ground biomass."*

ad 4) We have now given the details in the text and corrected the respective metadatafiles.

26. In the data files on biomass and LAI, I was also missing a column for comments. Sampling of vegetation frequently causes problems with dirty or wet leaves, biomass of weeds, occurrence of pests, and so on. This needs to be reported.

AGREE but ... we do not have this soft information and, thus, cannot report on it.

27. The data files report biomass and LAI per plot. In order to provide the opportunity to use the data with remote sensing data, please provide coordinates of the plots.

AGREE. Point well taken. However, we think the data is sufficiently included in the gis data model, in which the plot location is precisely documented.

28. The data file on phenology also reports on a replicate. Please explain that in the text.

AGREE. We added a sentence to specify that both phenology and plant height were measured on individual plants in the plots. We now added a sentence to give the details.

> *"In each plot of the research fields, observations of phenology and plant height on ten different plants were made and are reported as plot replicates in the files."*

29. Furthermore, plant height is included in this file but users would look for it in files for biomass or LAI. Please move it there.

DISAGREE. See comment above. Phenology and plant height measurements were performed on individual plants on a plot, so should be in the same file as they are directly linked in space and time, unlike LAI and biomass measurements. Admittedly, this was not clear in the text and we are convinced the sentence in response to comment 28 solves this.

Section 2.3.5: Based on the heading of this section, I was looking for GVF data in the dataset. This remained unsuccessful. The dataset only contains photos that can be used to derive GVF. This needs to be clear to readers. Therefore, this section can only be on photos. You may then describe how GVF can be derived from the photos. If I got something wrong and GVF is part of the dataset, the following has to be noted: The hint on the possibility to combine gfv with RapidEye is valuable for readers. However, the results of the study of Bohm et al. (2020) does not belong in this paper. Therefore, remove the sentences from P15L12 on. This section mentions the biological names of the crops. Because this is not relevant in the context of photos, these names should be removed here. They may be added where the crops are introduced or in the plant metadata file.

> AGREE. Correctly noted, the GVF is not part of the dataset, only the photos are. Thus, the first sentence of the paragraph is misleading. We have renamed and considerably shortened the paragraph as advised.

30. Section 2.4: Because this section is on compartments of the system for which other data has already been described above, the part on soil should be added in the soil section. The same applies to information on soil CN measurements, which are addressed here but described in section 2.5.
    The information in the plant section is a repetition of section 2.3.4, anyway, and can therefore be removed.

AGREED. We moved the soil part, as requested, deleted the plant part as requested.

31. Section 2.4 soil: I cannot find the files for the bare soil plots in for1695_data.zip.

There are none. The data is contained together with the vegetated plots. Our aim was to reduce the number of files. This was incorrect in the corresponding metadata and corrected by us.

32. In the second paragraph, information on manual tilling is given. However, it remains unclear, why this had to be done. Were the plots excluded from the management by the farmer? This would mean, that the management data does not apply to the plant data from these plots. Please explain in more detail. Because this information is of relevance for the plant data, mention it in the respective section.

AGREED. We reworked the chamber flux measurement section and merged it with the soil-atmosphere flux section 2.3.2. here we give details on how the bare soils were managed and what inputs there are to be expected. Here, we explicitly give details on the respiration measurements using the chamber method.

> *"Both types of soil respiration plots (bare and vegetated) were located close to the plots used for plant observations/biomass harvests. Additional bare-fallow plots were established in 2009, 2010 and 2012 (bare09, bare10, bare12, respectively) in the research fields. Plots were kept clear of vegetation during the experiment by manual weeding and periodic spot spraying of glyphosate (Monsanto Agrar, GmbH, Germany). The plots were tilled by hand in a way as to mimic mechanized tillage. In addition to plant residues, the vegetated plots received manure/slurry and mineral fertilizer as organic inputs from the farmers' field management.*

> *Since bare plots were maintained through multiple years of the experiment, the plots were located at the periphery of the fields to be outside of the EC footprint. Each of the three plots was allocated randomly within a third of the field outside the main EC footprint avoiding field edges, tractor pathways, and other non-representative spots. Soil surface $CO_2$ flux was measured via portable infrared gas analyzers (EGM-2 and EGM-4, PP Systems*

*Amesbury, Massachusetts, USA) with attached soil temperature probe and soil respiration chamber (Demyan et al., 2016; ). The soil respiration flux was 10 cm in diameter with an internal volume of 1171 cm−3. Fluxes were measured during the growing period of different years (2009, (n=5); 2010, (n>15); 2014, (n>3); 2015, (n=7) and an intensive approximately weekly campaign May 2012 to June 2013, (n=40)). Six replicate measurements were taken within each subplot (vegetated and bare fallow) during each measurement day. Measurement order of the plots was randomized each day to avoid time of day effects. Individual measurements which were greater than six times the yearly median value were removed as outliers."*

33. Section 2.5: Analogous to section 2.4, the information in this section should be combined with the other information on the same compartments (soil, plant).

AGREED. done. The original section 2.5 no longer exists. Now section 2.5. makes some statements to measurement errors and uncertainties.

34. Section 2.5.1: Soil organic C content is assigned the symbol TOC while in the data file it is SOC. From the text, it is not clear to me whether these are different measurements. The different devices (Vario-EL vs. MACRO Cube) seem to support this. Please rephrase and clarify. I suggest removing TOC from the text if it is not included in the data.

    The last line of the section mentions gravimetric estimation of SWC. Because this data is not included in the dataset, remove this information.

AGREED. done.

35. Section 2.5.2: Many studies report drying of vegetation samples at temperatures above 100 °C. Please cite a study that shows that 60 °C or only 28 °C is sufficient or report on the comparison of the dry weight before and after determination of residual water content.

DISAGREE. In this context it is only important that we state how we have done it. To determine the residual water content, indeed 105°C were used, as stated now more clearly (also in the metadata files).

36. All sections: In each section, refer the reader to the data files that contain the respective data.

AGREE.

DATAFILES:

37. Data: Using NAN as the missing data symbol can cause problems since software will read this as an error instead of a missing number. Missing values should be marked by a different code like e.g. "NA".

**\*sighs\*** ☺ AGREE. Changed from NAN to NA.

38. Metadata files: There are no column headers for columns 2 and 3. Please add headers for each table or mention in the beginning of each file. If files were formatted using spaces instead of tabs, they were readable independent of the tab-settings of a user's editor.

Nothing done. We checked if all data and metadata is machine readable (with R), and it is!

39. 00_metadata/research_sites_metadata.txt: Missing values are stated to be NAN but missing site names are marked by NN.

Nothing done. We now specify that NN means no name, and not

40. 00_metadata/management_metadata.txt: L28: What is meant by "Mark the large variability […]"?

No idea. Deleted the half sentence.

41. 00_metadata/soil_metadata.txt: for variable wp add "permanent wilting point" for users not familiar with pF values.

AGREED. Added definition of pF to the corresponding lines:
*"pF is defined as log10(| matric potential |), i.e. the decadal logarithm of the absolute value of matric potential"*

42. 00_metadata: names of site-specific soil metadata files include EC, which is misleading. Consider replacing "EC" by "site".

AGREED. Done

43. 05_management/fertilization/ fertilization.csv: contains three empty row at the end. The last line with data is assigned to site 5, is that right? The related Excel file contains additional sheets.

AGREED. Deleted

44. 05_management/cultivation/correction.txt: Correction history only needs to be shown for corrections after publication of the dataset

AGREED. deleted.

45. 05_management/soil_management/soil_management.csv: in the metadata, mention the Expert-N is a model

AGREED. done

46. 04_weather: double csv in filename

AGREED. corrected

47. 06_plant/biomass/not_publish_calc_biomass.csv: is this file meant to be included in the dataset?

AGREED. No, as its name suggests ;). Deleted.

48. 06_plant/lai/ and 06_plant/phenology/ contain a hidden file names ".plot", respectively

AGREED. deleted

49. 06_plant/plant_metadata.txt: the explanation of residual water content may be understood as if this was determined after drying at 60 °C.

AGREED. Corrected to 105°C.

50. **Technical comments and details**
51. P1L26: Here and on page 15, the name "rape" is used for Brassica napus. The rest of the text uses "rapeseed"

AGREE. Corrected in all cases.

52. P5L9: Use the same number of decimals for the coordinates of both sites.

AGREE. Corrected.

53. P5L9: The term "research sites" is not consistently used throughout the text

DISAGREE. We checked all instances of "research sites" in the text. it is always used for EC1-6, never for anything else. Thus, we cannot spot any ambiguous usage.

54. P5L14: […] in 2018, respectively, […]

DELETED. The sentence was deleted from section 2.1 and moved to the summary and conclusion. Also see comment 56.

55. Table 1, caption: three (?) research sites. Six! Mention, that this is the data from profile_data.csv.

AGREE. Corrected and included the suggestion.

56. P6L11: Which site is the one ongoing site (EC 1/2/3)?

AGREE. In the section *Summary and Conclusions* we now introduced a sentence to address this by:

*„One research site per region, (EC2 and EC4) are still fully operational, while the remaining sites were dismantled after completion of the project at the end of the growing season in 2018."*

57. P7L26: Sentence on coordinates: measurement period should be in parentheses not coordinates since these are the topic of the sentence.

AGREE. We solve this by rewriting the sentence (LL xxx):

*„Three EC stations (EC4, EC5, and EC6) were installed at fields with the respective areas of 8.7, 16.7, and 13.4 ha (Figure 2): EC4 (48°31'38.95"N, 9°46'9.73"E, 685 m a.s.l.) was in […]"*

58. Table 2: Can you mark the crop abbreviations in the legend below the table with the same colors?

AGREE partly. The idea is cool, but the colours are too light for it to be ensured that on a printout the words can still be read. We did not find an aesthetically pleasing solution, but have reformatted the table slightly to enhance readability.

59. P8L5, P8L13: Please correct "(c.f. 0)"

AGREE. The hyperlink broke. Hard coded to now read (c.f. section 2.3.4) in LL xxx and LL xxx

60. P8L24: Please add area reference to the 0.5 tons (per hectare?).

AGREE. done

61. P8L25: Can you give a reference for the lower yields of group B and C varieties?

DISAGREE partly: We do not state that yields of group B and C are lower. However, we now make reference to appendix B to reference the yields.
P8L26: Please refer the readers to the tables in appendix B in the text where district averages are mentioned.

AGREE. done (cf previous comment.

62. Table 3: Please explain the meaning of "group" in the caption. In the legend on crops, what does "E, A, B, C" mean?

AGREE. done. explanation included in the caption

63. Since the table contains fertilization information as the only element of a nutrient balance, the caption is misleading. Instead, yield could be mentioned in the caption.

AGREE. The caption now reads „*Table 3: Summary of field management, and nitrogen and organic matter input.*"

64. P10L3-12: This belongs to section 2.3.2. Information is partly repeated there.

AGREED. Deleted this entire section and added a sentence to 2.3.2 (LL xxx):

All six EC stations were equipped with the same equipment („Table 2), except for the number of soil sensors which was variable ( **Error! Reference source not found.**).

65. Table 4: There is no need to cite here without referring to additional relevant information on instrumentation in that paper.

AGREE. We reworded the caption to make it clear that the table was mostly published, elsewhere.

„Table 2: *The* instrumentation of the eddy-covariance and research s*tations* (Wizemann et al., 2015) *was used until 2018. O*ccasional changes to the general layout are detailed in the text *"*

66. Figure 3: "top" missing in caption. What does "SJ" refer to? Description of the data shown should be moved to the text.

AGREED. Well spotted. Amended.

67. P11L12f: The sign convention should also be mentioned in the metadata file for weather.

AGRED. Done. Also in the text.

68. P11L13f: This sentence seems to be related to Figure 3 but no reference to the figure is given. In addition, there is something missing before the comma.

AGREED. Reference to Figure 3 disappeared.

69. P11L16: Figure 5 rather belongs to section 2.3.3

DISAGREE. In Figure 5 (now Figure 4), measurements from several compartments are combined. It is best mentioned, where the data is first presented.

70. P12L29: There is more in the figures than just data coverage.

AGREE. This sentence is terrible! We have rewritten it to now read (LL xxx):
"By way of example, time series of $CO_2$ fluxes are presented in **Error! Reference source not found.**5 for EC1, which also shows presents the data coverage. Data for EC2-6 can be found in the supporting information."

71. P12L30: "Supplementary" instead of "supporting"

AGREED. changed as requested.

72. P14L9: Hukseflux

AGREED. Well spotted. Changed.

73. P14L13: There is no data on soil temperature or moisture in Table 3.

AGREED. Deleted reference.

74. P14L24: Please explain what "five extra plants" means. Why are they "extra"?

AGREED. Deleted the word Extra

75. Figure 6: The figure is not consistent with the data in for1695_data.zip. The same applies to the readme file in for1695_data_GVF.zip

AGREED. Sorry about this. Absolutely right. Thank you, and done-

76. P17L14: This applies to comparisons with the aim to find effects of the location only.

AGREED.

77. Tables 7ff: Consider unifying the beginning of the captions with or without "determined".

AGREED.

78. Terms: Please unify "matric potential" and "matrix potential". Both are used in the text or metadata files.

AGREED. Checked and done.

79. Tables in appendix B: Mention source of the data in the caption.

AGREED. This is very important. Thank you for the heads-up! We now added the reference also to the reference list: Statistisches Landesamt Baden-Württemberg. Regionaldaten - Land- und Forstwirtschaft - Ernte - Hektarerträge der Feldfrüchte (Landkreis ###). Accessible at: https://www.statistik-bw.de/SRDB/. Last accessed: 20 May2020.

80. Please check the tables in section 3 and the metadata files for consistency.

AGREED. Checked and done.

81. References for citations in metadata files (like Priesack, 2006 or Foken, 2006, van Genuchten, 1980, Ad-hoc-AG Boden, 2005) need to be given either in the metadata files or in the list of references in the text, or, maybe preferably as a service to the users, in both places.

AGREED. Excellent suggestion. Done!

Additional changes
1) The affiliation of Dr. Christian Troost was listed twice which was now corrected.
2) Figure 4 and Figure 5 are now Figure 5 and Figure 4, respectively. Changes in the text were made.
3) We made some minor edits to enhave readability and correct spelling.
4) The winterwheat yield in 2016 was erroneously reported as 6.6 t/ha. We corrected this to 7.7. Both in the Table 2, and in the data file.
5) We found an error in the plant_metadata.csv. The temperature for residual drying was given as 60°C. In fact it was 105°C. This was also corrected in table 14.
6) The numbering of Tables and Figures have changed.
7) We updated the information about the data access in the introduction by specifiyng:

„The data set was uploaded to the BONARES data centre (https://www.bonares.de/datacentre) *and is available* at https://doi.org/10.20387/bonares-a0qc-46jc."

---

## Author Comment (AC2)

General comments

The authors aim to provide a comprehensive dataset including multi-site, multi-crop measurements of the water, heat and carbon fluxes as well as the associated parameters in the Soil-Plant-Atmosphere-Continuum system in Germany. When the dataset is unique and complete, and be useful for the scientific community, the describing manuscript, however, is not well organized and is not easy to follow.

Dear Reviewer, thank you for the time and effort put into this review. Based on RC1, we have greatly tightened the structure of the manuscript, deleted redundancies, and made sure that we addressed some missing points, which we do not fully include here.
Most importantly, this addresses

a) Addition of missing text pieces, without which some parts were indeed not that clear
b) deleted section 2.4, and integrated the relevant parts to where the different environmental compartments are described, namely section 2.3.

With this, we are convinced to have enhanced readability greatly.

Specifically,

1. The manuscript focus too much on the description of the measurement detail (section 2). Nearly a half of the text is on the measurement detail and the site description, the description of the dataset in comparison is very simple. Surely, the measurement detail is very important. However, it can be presented in a better and clear way, such as in a table describing e.g., the specific devices, the measured method and the time duration for each parameter (an extended table 4). Besides, the sub-sections, such as field measurements (2.3), field sampling (2.4) and laboratory measurements (2.5) should be reorganized in a brief and clear way.

DISAGREE partly. Following the journal's scope is to attract "*articles [that] [...] may pertain to the planning, instrumentation, and execution of experiments or collection of data*" which is exactly the scope of our article. Further, the journal's scope is detailed that "a*ny interpretation of data is outside the scope of regular articles*". We are convinced that we have stringently abided by the scope.

We do agree, that the organisation of the section 2.3, 2.4, and 2.5 deserved improving. As detailed in responding to RC1, we have

1) removed section 2.4 and merged it with section 2.3. In section 2.3 we now directly give the details on field sampling methods, as well as on the measurement process for the soil respiration.
2) The introduction to section 2 (Material and Methods) now has a) a graphical schema to explain the system, and b) gives a more detailed description and chronological description of the entire section 2.

Thereby, we are convinced that this greatly enhances the readability, structure and overall clarity of the article.

2.  The authors mixed the data measurement and the data analysis (figures 3-5) together in section 2. It is better to present separately. The authors may also consider extending the data analysis section. Since the manuscript contains so many different data, it is useful for readers to briefly introduce the dynamic of the main parameters and the difference between the two sites.

DISAGREE partly. We just give very global descriptions of the sites as a guide to the readers. our intent is not to repeat any of the currently achieved data analyses.

3.  For this dataset, many parameters in relation with soil, plant and atmosphere properties were measured. I suggest to include a figure that demonstrates all the parameters measured from soil to atmosphere, including the specific parameter, the observed depth, the time duration and others. This would make the presentation more clearly. Note that this should be in more detail than the figure 1 in the text.

AGREE. Great idea!! We did this by including Figure 2

[Figure]

**Figure 1: Schema of the core of the measurement campaign at the research sites: 1 - soil profile characteristics, 2 - management and cultivation data (sowing date, harvest date, crop type and variety, fertilisation and pesticide application including amount and type), soil tillage, 3 - meteorological data (rain, air temperature at two meters height, and relative humidity), 4 - soil/biosphere-atmosphere fluxes using fully equipped eddy covariance stations for carbon, energy, and water vapor flux measurements, as well as wind speed and wind direction, 5 - soil state measurements including water content, temperature, and matric potential, the soil profile depth permitting, at 5 cm, 15 cm, 45 cm, 75 cm, 90 cm, 130 cm soil depth, and 6 – five plots per research site for carbon and nitrogen measurements integrated over depths of 0-30 cm, 30-60 cm, 60-90 cm, and 7 - plant performance also determined at the plots (phenology, height, and leaf area index, yield, above ground biomass, carbon and nitrogen in vegetative and generative biomass). Illustration by H. Vanselow (http://www.holgervanselow.de/).**

4. For the provided dataset (for1695_data), a description of the data quality should be provided. Also please check the format of the text in the 00_meatadata file, please provide more information for the calibration equation described in the ec1_soil_metadata file.

We understand that this point considers three different aspects.

1) data quality
2) format of the text in 00_metadata file
3) calibration described in the ec1_soil_metadata

ad 1) We have now included a sub-section on uncertainties (2.5).

ad 2) We checked.

ad 3) Done, as detailed in response to RC1

Specific comments

1. P4L15. The description of the dataset in the follow section is better in the same order with the summarize here. For example, the soil characteristics here is in the order of (iv) whereas it is presented firstly in the section 2.

AGREE and a very good point, too. Thanks! This list now follows exactly the order in the Material and Methods section.

2. P7L2. It is better to present Table 1 in the soil measurement section.
3. P11. The author may consider placing the (briefly) data analysis (figures 3-5) in a new data analysis section.

DISAGREE. Thanks for the idea. However, in the introduction and throughout the manuscript, we cite relating manuscript.

4. P14L15. Please complete the description.

AGREED. Done.